# NEUPL: NEURAL POPULATION LEARNING

**Siqi Liu**
University College London
DeepMind
`liusiqi@google.com`

**Luke Marris**
University College London
DeepMind
`marris@google.com`

**Daniel Hennes**
DeepMind
`hennes@google.com`

**Josh Merel**[*]
DeepMind
`jsmerel@gmail.com`

**Nicolas Heess**
DeepMind
`heess@google.com`

**Thore Graepel**[†]
University College London
`t.graepel@ucl.ac.uk`

## ABSTRACT

Learning in strategy games (e.g. StarCraft, poker) requires the discovery of diverse policies. This is often achieved by iteratively training new policies against existing ones, growing a policy population that is robust to exploit. This iterative approach suffers from two issues in *real-world* games: a) under finite budget, approximate best-response operators at each iteration needs truncating, resulting in under-trained good-responses populating the population; b) repeated learning of basic skills at each iteration is wasteful and becomes intractable in the presence of increasingly strong opponents. In this work, we propose Neural Population Learning (NeuPL) as a solution to both issues. NeuPL offers convergence guarantees to a population of *best*-responses under mild assumptions. By representing a population of policies within a single conditional model, NeuPL enables transfer learning across policies. Empirically, we show the generality, improved performance and efficiency of NeuPL across several test domains[1]. Most interestingly, we show that novel strategies become more accessible, not less, as the neural population expands.

The need for learning not one, but a population of strategies is rooted in classical game theory. Consider the purely cyclical game of *rock-paper-scissors*, the performance of individual strategies is meaningless as improving against one entails losing to another. By contrast, performance can be meaningfully examined *between* populations. A population consisting of pure strategies $\{rock, paper\}$ does well against a singleton population of $\{scissors\}$ because in the meta-game where both populations are revealed, a player picking strategies from the former can always beat a player choosing from the latter[2]. This observation underpins the unifying population learning framework of Policy Space Response Oracle (PSRO) where a new policy is trained to best-respond to a mixture over previous policies at each iteration, following a meta-strategy solver (Lanctot et al., 2017). Most impressively, Vinyals et al. (2019) explored the strategy game of StarCraft with a league of policies, using a practical variation of PSRO. The league counted close to a thousand sophisticated deep RL agents as the population collectively became robust to exploits.

Unfortunately, such empirical successes often come at considerable costs. Population learning algorithms with theoretical guarantees are traditionally studied in normal-form games (Brown, 1951; McMahan et al., 2003) where best-responses can be solved exactly. This is in stark contrast to real-world *Game-of-Skills* (Czarnecki et al., 2020) — such games are often temporal in nature, where best-responses can only be approximated with computationally intensive methods (e.g. deep RL). This has two implications. First, for a given opponent, one cannot efficiently tell apart *good*-responses that temporarily plateaued at local optima from globally optimal best-responses. As a result, approximate best-response operators are often truncated prematurely, according to hand-crafted schedules (Lanctot et al., 2017; Mcaleer et al., 2020). Second, real-world games often afford strategy-agnostic transitive

---

[*]Currently at Reality Labs, work carried out while at DeepMind.
[†]Work carried out while at DeepMind.
[1]See `https://neupl.github.io/demo/` for supplementary illustrations.
[2]This is formally quantified by Relative Population Performance, see Definition A.1 (Balduzzi et al., 2019).

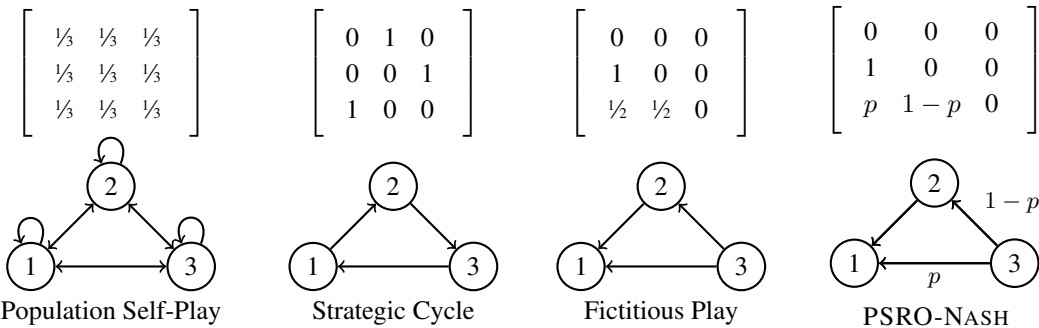

Figure 1: Popular population learning algorithms implemented as directed interaction graphs (**Bottom**), or equivalently, a set of meta-game mixture strategies $\Sigma \in \mathbb{R}^{3\times3} := \{\sigma_i\}_{i=1}^{3}$ (**Top**). A directed edge from $i$ to $j$ with weight $\sigma_{ij}$ indicates that policy $i$ optimizes against $j$, with probability $\sigma_{ij}$. Unless labeled, out edges from each node are weighted equally and their weights sum up to one.

skills that are pre-requisite to strategic reasoning. Learning such skills from scratch at each iteration in the presence of evermore skillful opponents quickly becomes intractable beyond a few iterations.

This iterative and isolated approach is fundamentally at odds with human learning. For humans, mastering diverse strategies often facilitates incremental strategic innovation and learning about new strategies does not stop us from revisiting and improving upon known ones (Caruana, 1997; Krakauer et al., 2006). In this work, we make progress towards endowing artificial agents with similar capability by extending population learning to real-world games. Specifically, we propose NeuPL, an efficient and general framework that learns and represents diverse policies in symmetric zero-sum games within a single conditional network, using the computational infrastructure of simple self-play (**Section 1.2**). Theoretically, we show that NeuPL converges to a sequence of iterative best-responses under certain conditions (**Section 1.3**). Empirically, we illustrate the generality of NeuPL by replicating known results of population learning algorithms on the classical domain of *rock-paper-scissors* as well as its partially-observed, spatiotemporal counterpart *running-with-scissors* (Vezhnevets et al., 2020) (**Section 2.1**). Most interestingly, we show that NeuPL enables transfer learning across policies, discovering exploiters to strong opponents that would have been inaccessible to comparable baselines (**Section 2.2**). Finally, we show the appeal of NeuPL in the challenge domain of MuJoCo Football (Liu et al., 2019) where players must continuously refine their movement skills in order to coordinate as a team. In this highly transitive game, NeuPL naturally represents a short sequence of best-responses without the need for a carefully chosen truncation criteria (**Section 2.4**).

# 1 METHODS

Our method is designed with two desiderata in mind. First, at convergence, the resulting population of policies should represent a sequence of iterative best-responses under reasonable conditions. Second, transfer learning can occur across policies throughout training. In this section, we define the problem setting of interests as well as necessary terminologies. We then describe NeuPL, our main conceptual algorithm as well as its theoretical properties. To make it concrete, we further consider *deep RL* specifically and offer two practical implementations of NeuPL for real-world games.

## 1.1 PRELIMINARIES

**Approximate Best-Response (ABR) in Stochastic Games**  We consider a symmetric zero-sum Stochastic Game (Shapley, 1953) defined by $(\mathcal{S}, \mathcal{O}, \mathcal{X}, \mathcal{A}, \mathcal{P}, \mathcal{R}, p_0)$ with $\mathcal{S}$ the state space, $\mathcal{O}$ the observation space and $\mathcal{X} : \mathcal{S} \to \mathcal{O} \times \mathcal{O}$ the observation function defining the (partial) views of the state for both players. Given joint actions $(a_t, a'_t) \in \mathcal{A} \times \mathcal{A}$, the state follows the transition distribution $\mathcal{P} : \mathcal{S} \times \mathcal{A} \times \mathcal{A} \to \Pr(\mathcal{S})$. The reward function $\mathcal{R} : \mathcal{S} \to \mathbb{R} \times \mathbb{R}$ defines the rewards for both players in state $s_t$, denoted $\mathcal{R}(s_t) = (r_t, -r_t)$. The initial state of the environment follows the distribution $p_0$. In a given state $s_t$, players act according to policies $(\pi(\cdot|o_{\leq t}), \pi'(\cdot|o'_{\leq t}))$. Player $\pi$ achieves an expected return of $J(\pi, \pi') = \mathbb{E}_{\pi,\pi'}[\sum_t r_t]$ against $\pi'$. Policy $\pi^*$ is a *best response* to $\pi'$

if $\forall \pi, J(\pi^*, \pi') \geq J(\pi, \pi')$. We define $\hat{\pi} \leftarrow \text{ABR}(\pi, \pi')$ with $J(\hat{\pi}, \pi') \geq J(\pi, \pi')$. In other words, an ABR operator yields a policy $\hat{\pi}$ that does no worse than $\pi$, in the presence of an opponent $\pi'$.

**Meta-game Strategies in Population Learning** Given a symmetric zero-sum game and a set of $N$ policies $\Pi := \{\pi_i\}_{i=1}^N$, we define a normal-form meta-game where players' $i$-th action corresponds to executing policy $\pi_i$ for one episode. A meta-game strategy $\sigma$ thus defines a probability assignment, or an action profile, over $\Pi$. Within $\Pi$, we define $\mathcal{U} \in \mathbb{R}^{N \times N} \leftarrow \text{EVAL}(\Pi)$ to be the expected payoffs between pure strategies of this meta-game or equivalently, $\mathcal{U}_{ij} := J(\pi_i, \pi_j)$ in the underlying game. We further extend the ABR operator of the underlying game to mixture policies represented by $\sigma$, such that $\hat{\pi} \leftarrow \text{ABR}(\pi, \sigma, \Pi)$ with $\mathbb{E}_{\pi' \sim P(\sigma)}[J(\hat{\pi}, \pi')] \geq \mathbb{E}_{\pi' \sim P(\sigma)}[J(\pi, \pi')]$. Finally, we define $f : \mathbb{R}^{|\Pi| \times |\Pi|} \to \mathbb{R}^{|\Pi|}$ to be a meta-strategy solver (MSS) with $\sigma \leftarrow f(\mathcal{U})$ and $\mathcal{F} : \mathbb{R}^{N \times N} \to \mathbb{R}^{N \times N}$ a meta-graph solver (MGS) with $\Sigma \leftarrow \mathcal{F}(\mathcal{U})$. The former formulation is designed for iterative optimization of approximate best-responses as in Lanctot et al. (2017) whereas the latter is motivated by concurrent optimization over a set of population-level objectives as in Garnelo et al. (2021). In particular, $\Sigma \in \mathbb{R}^{N \times N} := \{\sigma_i\}_{i=1}^N$ defines $N$ population-level objectives, with $\pi_i$ optimized against the mixture policy represented by $\sigma_i$ and $\Pi$. As such, $\Sigma \in \mathbb{R}^{N \times N}$ corresponds to the adjacency matrix of an interaction graph. Figure 1 illustrates several commonly used population learning algorithms defined by $\Sigma$ or equivalently, their interaction graphs.

## 1.2 NEURAL POPULATION LEARNING

We now present NeuPL and contrast it with Policy-Space Response Oracles (PSRO, Lanctot et al. (2017)) which similarly focuses on population learning with approximate best-responses by RL.

---

**Algorithm 1** Neural Population Learning (Ours)

1: $\Pi_\theta(\cdot|s, \sigma)$      ▷ Conditional neural population net.
2: $\Sigma := \{\sigma_i\}_{i=1}^N$      ▷ Initial interaction graph.
3: $\mathcal{F} : \mathbb{R}^{N \times N} \to \mathbb{R}^{N \times N}$      ▷ Meta-graph solver.
4: **while** true **do**
5:    $\Pi_\theta^\Sigma \leftarrow \{\Pi_\theta(\cdot|s, \sigma_i)\}_{i=1}^N$      ▷ Neural population.
6:    **for** $\sigma_i \in \text{UNIQUE}(\Sigma)$ **do**
7:      $\Pi_\theta^{\sigma_i} \leftarrow \Pi_\theta(\cdot|s, \sigma_i)$
8:      $\Pi_\theta^{\sigma_i} \leftarrow \text{ABR}(\Pi_\theta^{\sigma_i}, \sigma_i, \Pi_\theta^\Sigma)$      ▷ Self-play.
9:    $\mathcal{U} \leftarrow \text{EVAL}(\Pi_\theta^\Sigma)$      ▷ (Optional) if $\mathcal{F}$ adaptive.
10:    $\Sigma \leftarrow \mathcal{F}(\mathcal{U})$      ▷ (Optional) if $\mathcal{F}$ adaptive.
11: **return** $\Pi_\theta, \Sigma$

**Algorithm 2** PSRO (Lanctot et al., 2017)

1: $\Pi := \{\pi_0\}$      ▷ Initial policy population.
2: $\sigma \leftarrow \text{UNIF}(\Pi)$      ▷ Initial meta-game strategy.
3: $f : \mathbb{R}^{|\Pi| \times |\Pi|} \to \mathbb{R}^{|\Pi|}$      ▷ Meta-strategy solver.
4:
5: **for** $i \in [[N]]$ **do**      ▷ N-step ABR.
6:    Initialize $\pi_{\theta_i}$.
7:    $\pi_{\theta_i} \leftarrow \text{ABR}(\pi_{\theta_i}, \sigma, \Pi)$
8:    $\Pi \leftarrow \Pi \cup \{\pi_{\theta_i}\}$
9:    $\mathcal{U} \leftarrow \text{EVAL}(\Pi)$      ▷ Empirical payoffs.
10:    $\sigma \leftarrow f(\mathcal{U})$
11: **return** $\Pi$

---

NeuPL deviates from PSRO in two important ways. First, NeuPL suggests concurrent and continued training of all *unique* policies such that no good-response features in the population prematurely due to early truncation. Second, NeuPL represents an entire population of policies via a shared conditional network $\Pi_\theta(\cdot|s, \sigma)$ with each policy $\Pi_\theta(\cdot|s, \sigma_i)$ conditioned on and optimised against a meta-game mixture strategy $\sigma_i$, enabling transfer learning across policies. This representation also makes NeuPL general: it delegates the choice of *effective* population sizes $|\text{UNIQUE}(\Sigma)| \leq |\Sigma| = N$ to the meta-graph solver $\mathcal{F}$ as $\sigma_i = \sigma_j$ implies $\Pi_\theta(\cdot|s, \sigma_i) \equiv \Pi_\theta(\cdot|s, \sigma_j)$ (cf. Section 2.1). Finally, NeuPL allows for cyclic interaction graphs, beyond the scope of PSRO. We discuss the generality of NeuPL in the context of prior works in further details in Appendix D.

**N-step Best-Responses via Lower-Triangular Graphs** A popular class of population learning algorithms seeks to converge to a sequence of $N$ iterative best-responses where each policy $\pi_i$ is a best-response to an opponent meta-game strategy $\sigma_i$ with support over a subset of the policy population $\Pi_{<i} = \{\pi_j\}_{j<i}$. In NeuPL, this class of algorithms are implemented with meta-graph solvers that return *lower-triangular* adjacency matrices $\Sigma$ with $\Sigma_{i \leq j} = 0$. Under this constraint, $\sigma_0$ becomes a zero vector, implying that $\Pi_\theta(\cdot|s, \sigma_0)$ does not seek to best-respond to any policies. Similar to the role of initial policies $\{\pi_0\}$ in PSRO (Algorithm 2), $\Pi_\theta(\cdot|s, \sigma_0)$ serves as a starting point for the sequence of N-step best-responses and any fixed policy can be used. We note that this property further allows for incorporating pre-trained policies in NeuPL, as we discuss in Appendix D.1.

---

**Algorithm 3** A meta-graph solver implementing PSRO-NASH.

---

1: **function** $\mathcal{F}_{\text{PSRO-N}}(\mathcal{U})$                        $\triangleright \mathcal{U} \in \mathbb{R}^{N \times N}$ the empirical payoff matrix.
2:     Initialize meta-game strategies $\Sigma \in \mathbb{R}^{N \times N}$ with zeros.
3:     **for** $i \in \{1, \ldots, N-1\}$ **do**
4:        $\Sigma_{i+1,1:i} \leftarrow \text{SOLVE-NASH}(\mathcal{U}_{1:i,1:i})$        $\triangleright$ LP Nash solver, see Shoham & Leyton-Brown (2008).
5:     **return** $\Sigma$

---

One prominent example is PSRO-NASH, where $\pi_i$ is optimized to best-respond to the Nash mixture policy over $\Pi_{<i}$. This particular meta-graph solver is shown in Algorithm 3.

### 1.3 CONVERGENCE TO N-STEP BEST-RESPONSES VIA NEUPL

Under certain assumptions on the best-response operator, interaction graph, and meta-graph solver (MGS) we can construct proofs that NeuPL converges to an $N$-step best-response. We introduce the term *grounded* (Section C) to refer to interaction graphs and MGS that have a structure that imposes convergence to a unique set of policies. Certain interaction graphs are grounded, in particular, lower-triangular graphs are one such class which describe an $N$-step best response. In addition, certain MGSs are grounded, in particular, ones that operate on the sub-payoff and output a lower-triangular interaction graph, $\mathcal{F} : \mathcal{U}_{<i,<i} \to \Sigma_{i,<i}$. The lower-triangular maximum entropy Nash equilibrium (MENE) is one such grounded MGS. Therefore with sufficiently large $N$, NeuPL will converge to a normal-form Nash equilibrium. See Section C for the full definitions, theorems and proofs.

### 1.4 NEURAL POPULATION LEARNING BY RL

We now define the discounted return maximized by $\Pi_\theta(\cdot|o_{\leq t}, \sigma_i)$ in Equation 1. We denote $P(\sigma_i)$ as the probability distribution over policy $i$'s opponent identities $\sigma_j \in \{\sigma_1, \ldots, \sigma_N\}$. Intuitively, each policy is maximizing its expected returns in the underlying game under a double expectation: the first is taken over its opponent distribution, with $\sigma_j \sim P(\sigma_i)$ and the second taken under the game dynamics partly defined by the pair of policies $(\Pi_\theta(\cdot|o_{\leq t}, \sigma_i), \Pi_\theta(\cdot|o'_{\leq t}, \sigma_j))$.

$$J_{\sigma_i} = \underset{\sigma_j \sim P(\sigma_i)}{\mathbb{E}} \Big[ \underset{a \sim \Pi_\theta(\cdot|o_{\leq t}, \sigma_i), a' \sim \Pi_\theta(\cdot|o'_{\leq t}, \sigma_j)}{\mathbb{E}} \big[ \sum_t r_t \gamma^t \big] \Big] \tag{1}$$

To optimize $\Pi_\theta^\Sigma$ by RL, we jointly train an opponent-conditioned action-value[3] function, approximating the expected return of choosing an action $a_t$ given an observation history $o_{\leq t}$, following $\Pi_\theta(\cdot|o_{\leq t}, \sigma_i)$ thereafter in the presence of $\Pi_\theta(\cdot|o'_{\leq t}, \sigma_j)$, denoted by $Q(o_{\leq t}, a_t, \sigma_i, \sigma_j) = \mathbb{E}_{\Pi_\theta(\cdot|o, \sigma_i), \Pi_\theta(\cdot|o', \sigma_j)}[\sum_{\tau=t}^{t+T} \gamma^{\tau-t} r_\tau | o_{\leq t}, a_t]$ with $\gamma$ the discount factor. In the case of ABR by deep RL, we could additionally approximate the expected payoffs matrix $\mathcal{U}$ by learning a payoff estimator $\phi_\omega(\sigma_i, \sigma_j)$ minimizing the loss $\mathcal{L}_{ij} = \mathbb{E}_{o \sim \mathcal{D}} \big[ (\phi_\omega(\sigma_i, \sigma_j) - \mathbb{E}_{a \sim \Pi_\theta(\cdot|o, \sigma_i)}[Q_\theta(o, a, \sigma_i, \sigma_j)])^2 \big]$ where the expectation is taken over the state visitation distribution $\mathcal{D}$ defined by the pair of policies and the environment dynamics $\mathcal{P}$. In other words, $\phi_\omega(\sigma_i, \sigma_j)$ approximates the expected return of $\Pi_\theta(\cdot|o_{\leq}, \sigma_i)$ playing against $\Pi_\theta(\cdot|o'_{<}, \sigma_j)$. By connecting payoff matrix $\mathcal{U}$ to the learned $Q$ function, we can evaluate $\mathcal{U}$ efficiently, without explicitly evaluating all policies at each iteration.

Finally, we propose Algorithm 4 in the setting where the meta-graph solver $\Sigma^{\text{const}} \leftarrow \mathcal{F}(\mathcal{U})$ is a *constant* function and extends it to Algorithm 5 where the meta-graph solver is adaptive in $\mathcal{U}$. For instance, population learning algorithms such as Fictitious Play (Brown, 1951) implement static interaction graphs while algorithms such as PSRO (Lanctot et al., 2017) rely on adaptive MGS.

## 2 EXPERIMENTS

In this section, we validate different contributions of NeuPL across several domains. First, we verify the generality of NeuPL from two aspects: **a)** NeuPL recovers expected results of existing population learning algorithms (Brown, 1951; Heinrich et al., 2015; Lanctot et al., 2017) on the

---

[3]We present the case for an action-value function but a value function could be used instead.

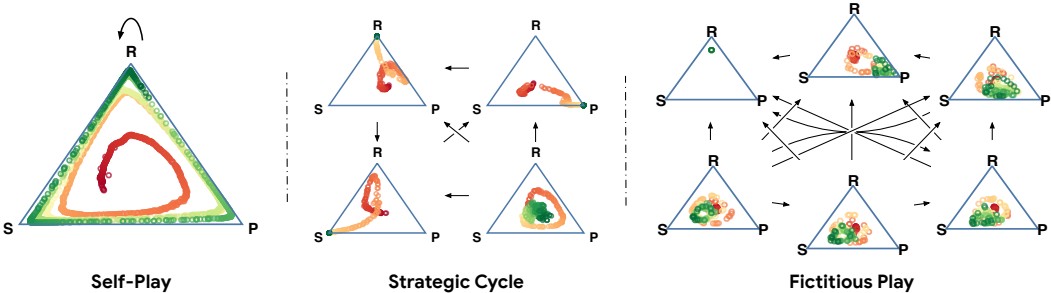

Figure 2: Neural Population Learning in *rock-paper-scissors* induced by static interaction graphs. Policy distributions are colored by training iteration from red (earliest) to green (latest). **(Left)** learning by self-play. **(Middle)** a neural population of 4 strategies exploring a strategic cycle. **(Right)** a neural population of 6 strategies iteratively best respond to the average previous strategies.

classical game of *rock-paper-scissors* where we can visualize the learned policy population over time and; **b)** NeuPL generalises to the spatiotemporal, partially observed strategy game of *running-with-scissors* (Vezhnevets et al., 2020), where players must infer opponent behaviours through tactical interactions. Second, we show that NeuPL induces skill transfer across policies, enabling the discovery of exploiters to strong opponents that would have been out-of-reach otherwise. This property translates to improved efficiency and performance compared to PSRO-NASH baselines, even under favorable conditions. Lastly, we show that NeuPL scales to the large-scale Game-of-Skills of MuJoCo Football (Liu et al., 2019) where a concise sequence of best-responses are learned, reflecting the prominent transitive skill dimension of the game.

In all experiments, we use Maximum A Posterior Optimization (MPO, Abdolmaleki et al. (2018)) as the underlying RL algorithm, though any alternative can be used instead. Similarly, any conditional architecture can be used to implement $\Pi_\theta^\Sigma$. Our specific proposal reflects the spinning-top geometry (Czarnecki et al., 2020) so as to encourage positive transfers across polices. Further discussions on the network design is available in Appendix B.2.

## 2.1 Is NEUPL General?

**Static Interaction Graphs**   Figure 2 illustrates the effect of NEUPL-RL-STATIC implementing popular population learning algorithms in the purely cyclical game of *rock-paper-scissors*. Figure 2 (Left) shows that learning by self-play leads to the policy cycling through the strategy space indefinitely, as expected in such games (Balduzzi et al., 2019). By contrast, Figure 2 (Middle) shows the effect of a specialized graph that encourages the discovery of a strategic cycle as well as a final strategy that trains against the others equally. As a result, we obtain a population that implements the pure strategies of the game as well as an arbitrary strategy. This final strategy needs not be the Nash of the game as any strategy would achieve a return of zero. Finally, Figure 2 (Right) recovers the effect of Fictitious Play (Brown, 1951; Heinrich et al., 2015) where players at each iteration optimize against the "average" previous players. We initialize the initial sink strategy to be exploitable, heavily biased towards playing rock[4]. The resulting population, represented by $\{\Pi_\theta(a|o_\leq, \sigma_i)\}_{i=1}^6$, learned to execute 6 strategies, starting with "pure-rock" which is followed by its best-response "pure-paper", with subsequent strategies gravitating towards the Nash equilibrium (NE) of this game.

**Adaptive Interaction Graphs**   Figure 3 illustrates that NEUPL-RL-ADAPTIVE with $\mathcal{F}_{\text{PSRO-N}}$ recovers the expected result of PSRO-NASH in *rock-paper-scissors*. Specifically, the first three strategies gradually form a strategic cycle and converge to the pure strategies of the game. As the cycle starts to form, the final strategy, best-responding to the NE over previous ones, shifted to optimize against a mixture of pure strategies. These results further highlight an attractive property of NeuPL — the number of distinct strategies represented by the neural population grows dynamically in accordance with the meta-graph solver (comparing $\Sigma$ at epoch 0 and 71). In particular, a distinct

---

[4]This choice is important: had the sink strategy been initialized to be the Nash mixture strategy, subsequent strategies would be "uninteresting" as no strategy can improve upon the sink strategy.

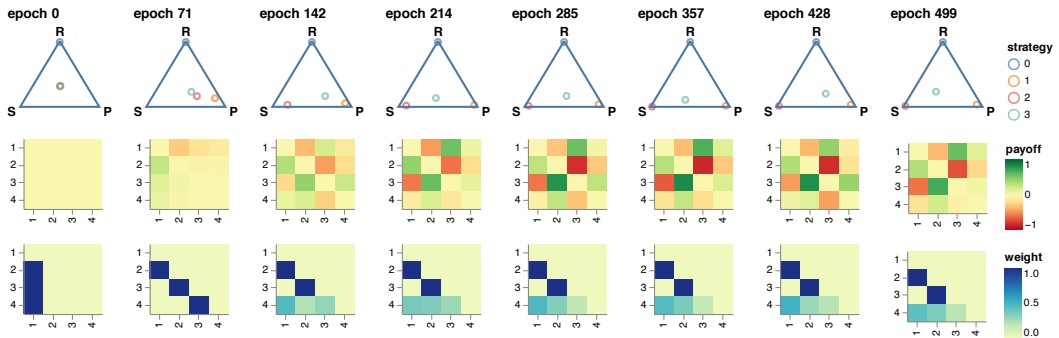

Figure 3: NeuPL in *rock-paper-scissors* induced by an adaptive interaction graph implementing PSRO-NASH. An epoch lasts 10 iterations. **(Top)** strategy space explored by the neural population of strategies over time. **(Middle)** the learned payoff estimates between the population of strategies. **(Bottom)** Adjacency matrices representing interaction graphs as responses to payoff matrices.

objective for strategy $i + 1$ is introduced if and only if $\Pi_\theta(\cdot|o_\leq, \sigma_i)$ gains support in the NE over $\Pi_\theta^{\Sigma \leq i}$. Unlike prior works (Lanctot et al., 2017; Mcaleer et al., 2020), the effective population size is driven by the meta-graph solver, rather than handcrafted truncation criteria. This is particularly appealing in real-world games, where we cannot efficiently determine if a policy has converged to a best-response or temporarily plateaued at local optima. In NeuPL, one needs to specify the maximum number of policies $N$ to be represented in the neural population, yet the population need not optimize $N$ distinct objectives at the start, nor is it required to converge to $N$ distinct policies at convergence. The number of distinct polices represented in the neural population is a function of the strategic complexity of the game, the capacity of the neural network, the effectiveness of the ABR operator and the nature of the meta-graph solver. We recall this property in *running-with-scissors* and in MuJoCo Football, which afford varying degrees of strategic complexities.

**Stochastic Games** *running-with-scissors* (Vezhnevets et al., 2020) extends *rock-paper-scissors* to the spatiotemporal and partially-observed setting. Using first-person observations of the game (a 4x4 grid in front of the agent), each player collects resources representing "rock", "paper" and "scissors" so as to counter its opponent's hidden inventory. At the end of an episode or when players confront each other through "tagging", players compare their inventories and receive rewards accordingly. To do well, one must infer opponent behaviours from its partial observation history $o_{\leq t}$ — if "rock"s went missing, then the opponent may be collecting them; if the opponent ran past "scissors", then it may not be interested in it. We describe the environment in details in Appendix B.1. Figure 4 shows that NeuPL with $\mathcal{F}_{\text{PSRO-N}}$ leads to a population of sophisticated policies. As before, we set the initial sink policy to be exploitable and biased towards picking up "rock"s. Early in training, we note that the first three policies of the neural population implement the pure-resource policies of "rock", "paper" and "scissors" respectively, as evidenced by their relative payoffs. In contrast to *rock-paper-scissors*, the mixture of pure-resource policies is exploitable in the sequential setting, where the player can observe its opponent before implementing a counter strategy. Indeed, policy $\Pi_\theta(\cdot|o_\leq, \sigma_4)$ observes and exploits, beating the mixture policies at epoch 680. Following $\mathcal{F}_{\text{PSRO-N}}$, $\Pi_\theta(\cdot|o_\leq, \sigma_5)$ updates its objective to focus solely on this newfound NE over $\Pi_\theta^{\Sigma < 5}$, developing a deceptive counter strategy.

Figure 5 (Left) quantitatively verify that NeuPL implementing $\mathcal{F}_{\text{PSRO-N}}$ indeed induces a policy population that becomes robust to adversarial exploits as the population expands. To this end, we compare independent policy populations by their Relative Population Performance (RPP[5], Appendix A.1) across 4 independent NeuPL experiments with different maximum population sizes. As expected, we observe that neural populations representing more best-response iterations are less exploitable. Additionally, a population size greater than 8 has limited impact on learning, both in terms of marginal exploitability benefits and rate of improvement against smaller populations (shown in blue and orange). This further mitigate the concern of using a larger maximum population size than necessary. In fact, Figure 5 (Right) shows that the effective population sizes $|\text{UNIQUE}(\Sigma)|$ plateau at 12 across maximum neural population sizes. We hypothesise that this is due to the increased difficulty in

---

[5]A negative RPP($\mathfrak{B}, \mathfrak{D}$) implies that all mixture policies in $\mathfrak{B}$ are exploitable by a mixture policy in $\mathfrak{D}$.

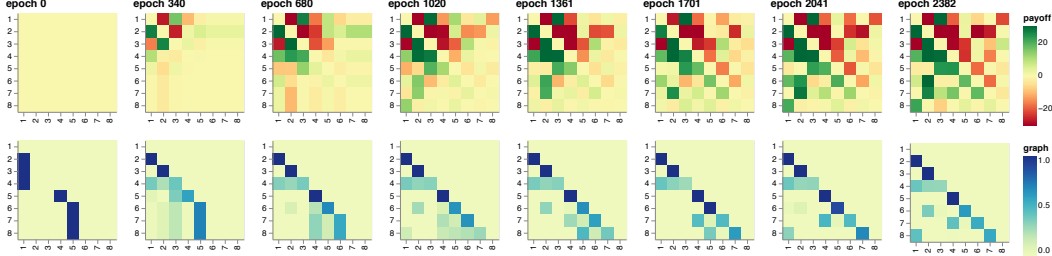

Figure 4: A NeuPL population developing an increasingly sophisticated set of diverse policies in *running-with-scissors*. The interaction graph is updated every 1,000 gradient updates (an epoch). **(Top)** the learned payoff estimates $\mathcal{U}$ between the neural population of policies as training progresses. **(Bottom)** interaction graphs $\Sigma \leftarrow \mathcal{F}_{\text{PSRO-N}}(\mathcal{U})$ as a response to the corresponding payoff matrix.

discovering reliable exploiter beyond 12 iterations in this domain — $\mathcal{F}_{\text{PSRO-N}}$ forms a curriculum where new objectives are introduced if and only if the induced meta-game contains novel meta-game strategies worth exploiting, regardless of the maximum population size specified. We emphasize that the only variable across these independent runs is their respective maximum neural population size.

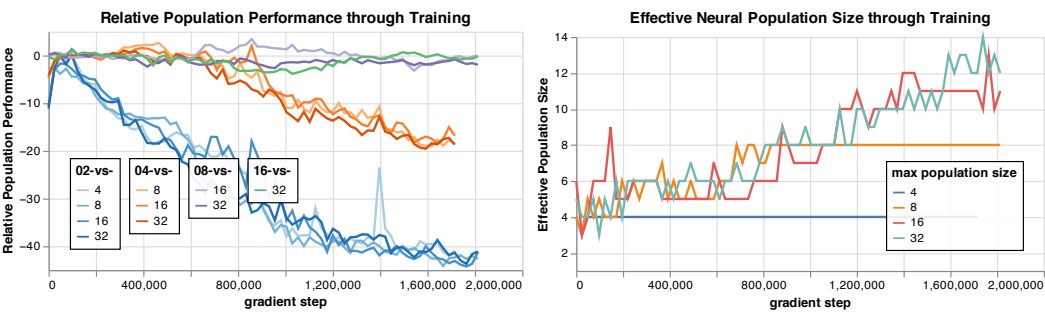

Figure 5: **(Left)** Relative Population Performance comparing independent NeuPL runs of different maximum population sizes. **(Right)** Effective population size over time, driven by $\mathcal{F}_{\text{PSRO-N}}$.

## 2.2 DOES NEUPL ENABLE TRANSFER LEARNING ACROSS POLICIES?

In contrast to prior works that train new policies iteratively *from scratch*, NeuPL represents diverse policies with explicit parameter sharing. Figure 6 compares the two approaches and illustrates the significance of transfer learning across iterations. Specifically, we verify that the shared representation learned by training against fewer, weaker policies early in training, facilitates the learning of exploiters to stronger, previously unseen opponents. To this end, a set of randomly initialized MPO agents with partially transferred parameters from different epochs of the experiment shown in Figure 4 are trained against fixed mixture policies defined by $\Pi_{\theta^*}^{\Sigma^*}$, with $\theta^*$ and $\Sigma^* = \mathcal{F}_{\text{PSRO-N}}(\mathcal{U}^*)$ obtained at epoch 1,200 of the same experiment. In other words, the objective for each agent is to beat a fixed NE over pre-trained policies $\{\Pi_{\theta^*}(a|o_{\leq}, \sigma_k^*)\}_{k=1}^n$, with a specific $n$. Figure 6 shows the learning progressions of the agents for $n \in \{2, 4, 7\}$, with transferred parameters taken at epoch 0 (red) upto 1,000 (green).

Against an easily exploitable opponent mixture (NE over the first two pure-resource policies), an agent with randomly initialized parameters (red) remains capable of learning an effective best response, albeit at a slower pace. This difference becomes much more apparent against competent mixture policies that execute sophisticated strategies (NE over the first 4 or 7 policies) — the randomly initialized agent failed to counter its opponent despite prolonged training while agents with partially transferred parameters successfully identified exploits, leveraging effective representation of the environment dynamics and of diverse opponent strategies. By transferring skills that support sophisticated strategic decisions across iterations, NeuPL enables the discovery of novel policies that are inaccessible to a randomly initialized agent. In other words, learning incremental best responses

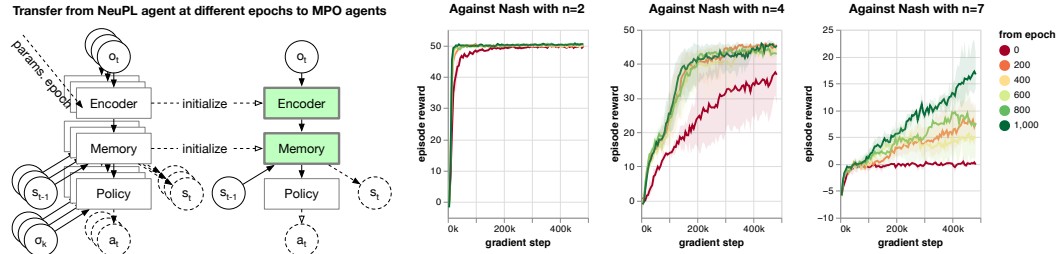

Figure 6: Learning progression of exploiters against incremental Nash mixture policies obtained via NeuPL training (as shown in Figure 4). The red curve corresponds to learning an exploiter from random initialization fully and the green curve corresponds to transferring encoder and memory components from the trained NeuPL network at epoch 1,000. Each experiment is repeated five times.

becomes easier, not harder, as the population expands. This is particularly attractive in games where strategy-agnostic skill learning is challenging in itself (Vinyals et al., 2019; Liu et al., 2021).

## 2.3 DOES NEUPL OUTPERFORM COMPARABLE BASELINES?

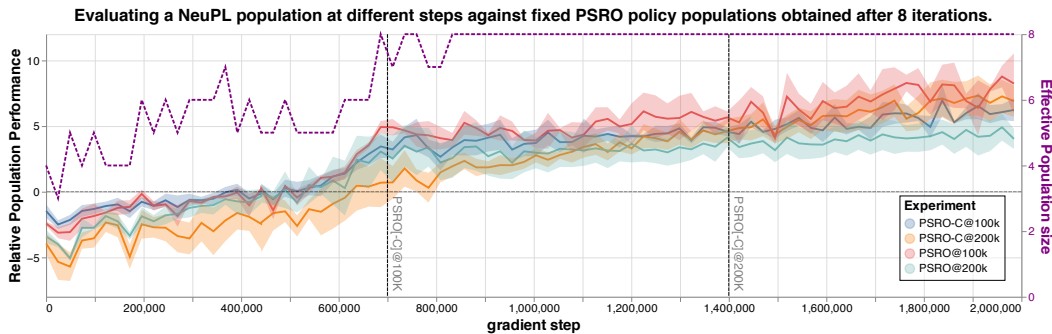

Figure 7: Relative Population Performance between a NeuPL population and policy populations obtained in PSRO baselines. Each PSRO variant is repeated over 3 trials, shown in shades.

We compare a NeuPL population implementing $\mathcal{F}_{\text{PSRO-N}}$ to 4 comparable baselines implementing variants of PSRO. Since PSRO does not prescribe a truncation criteria at each iteration, we investigate PSRO baselines with 100k and 200k gradient steps per iteration respectively. Further, we consider the effect of continued training across iterations, by initializing new policies with the policy obtained at the end of the preceding iteration instead of random initialization. We refer to this continued variant as PSRO-C. All PSRO populations are initialized with the same initial policy used for the NeuPL population. Figure 7 illustrates the quantitative benefits of NeuPL, measuring RPP between a NeuPL population of maximum population size of 8 against the *final* population of 8 policies obtained via PSRO after 7 iterations. The vertical dashed lines indicate that both the NeuPL population and the PSRO population have cumulatively undergone identical amount of training to allow for fair comparison. In purple, we show the effective population size of the NeuPL population which has been shown previously in Figure 5. We make the following observations: i) with a population size of 8, the NeuPL population successfully exploits PSRO baselines representing an equal number of policies, even if the latter performed twice as many gradient updates; ii) the increase in RPP coincides with an increase in the effective population size, from 5 to 8, reaching the maximum number of distinct policies that this NeuPL population can represent; iii) the amount of training each PSRO generation received has limited impact on the robustness of policy populations at convergence. This corroborates our observations in Figure 6, where the agent (in red) failed to exploit strong opponents despite continued training. Interestingly, PSRO-C proves equally exploitable. We hypothesize that the learned policies failed to develop reusable representations that can support diverse strategic decisions. Details of the PSRO baselines across 3 seeds are available in Appendix B.4, demonstrating the strategic complexities captured by the PSRO baseline populations.

### 2.4 Does NeuPL Scale to Highly Transitive Game-of-Skills?

If a game is purely transitive, all policies share the same best-response policy. In this case, self-play offers a natural curriculum that efficiently converges to this best-response (Balduzzi et al., 2019). Nevertheless, this approach is infeasible in real-world games as one cannot rule out strategic cycles in the game without exhaustive policy search. The MuJoCo Football domain (Liu et al., 2019) is one such example: it challenges agents to continuously refine their motor control skills while coordinated team-play intuitively suggests the presence of strategic cycles. In such games, PSRO is a challenging proposal as it requires carefully designed truncation criteria. If an iteration terminates prematurely due to temporary plateaus in performance, *good*-responses are introduced and convergence is slowed unnecessarily; if iterations rarely terminate, then the population may unduly delay the representation of strategic cycles. In such games, NeuPL offers an attractive proposal that retains the ability to capture strategic cycles, but also falls back to self-play if the game appears transitive.

Figure 8 shows the learning progression of NeuPL implementing $\mathcal{F}_{\text{PSRO-N}}$ in this domain, starting with a sink policy that exerts zero torque on all actuators. We observe that $\Pi_\theta(\cdot|o_\leq, \sigma_2)$ exploits the sink policy by making rapid, long-range shots which is in turn countered by $\Pi_\theta(\cdot|o_\leq, \sigma_3)$, that intercepts shots and coordinates as a team to score. Impressively, the off-ball blue player learned to blocked off defenders, creating scoring opportunities for its teammate. With the interaction graph focused on the lower diagonal elements, this training regimes closely matches that of self-play.

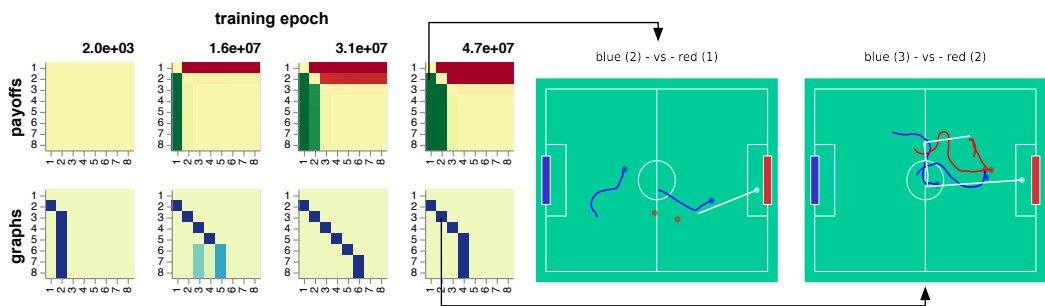

Figure 8: NeuPL in 2-vs-2 MuJoCo Football implementing $\mathcal{F}_{\text{PSRO-N}}$. The red/blue/white traces correspond to the trajectories of red/blue players and the ball respectively.

## 3 Related Work

Prior works attempted at making population learning scalable, motivated by similar concerns as ours. Mcaleer et al. (2020) proposed pipeline PSRO (P2SRO) which learns iterative best-responses concurrently in a staggered, hierarchical fashion. P2SRO offers a principled way to make use of additional computation resources while retaining the convergence guarantee of PSRO. Nevertheless, it does not induce more efficient learning per unit of computation cost, with basic skills re-learned at each iteration albeit asynchronously. In contrast, Smith et al. (2020a) focused on the lack of transfer learning across best-response iterations and proposed "Mixed-Oracles" where knowledge acquired over previous iterations is accumulated via an ensemble of policies. In this setting, each policy is trained to best-respond to a meta-game *pure* strategy, rather than a *mixture* strategy as suggested by the meta-strategy solver. To approximately re-construct a best-response to the desired mixture strategy, *Q-mixing* (Smith et al., 2020b) re-weights expert policies, instead of retraining a new policy. In comparison, NeuPL enables transfer while optimising Bayes-optimal objectives directly.

## 4 Conclusion and Future Work

We proposed an efficient, general and principled framework that learns and represents strategically diverse policies in real-world games within a single conditional-model, making progress towards scalable policy space exploration. In addition to exploring suitable technique from the multi-task, continual learning literature, going beyond the symmetric zero-sum setting remain interesting future works, too, as discussed in Appendix D.

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

# A  METHODS

## A.1  POPULATION LEARNING ALGORITHMS AS INTERACTION GRAPHS

Figure 1 shows several population learning algorithms represented as directed interaction graphs (Garnelo et al., 2021). In particular, Policy Space Response Oracle with a Nash meta-game solver (PSRO-NASH) (Lanctot et al., 2017) proposes to learn a best-response $\pi_i$ to the Nash equilibrium over policies $\Pi_{<i}$ at each iteration, resulting in an adaptive interaction graph. Specifically, the out-edges of node $i + 1$ are weighted according to the Nash equilibrium over the first $i$ policies (e.g. shown as $(1 - p, p)$ for node 3 in Figure 1).

## A.2  RELATIVE POPULATION PERFORMANCE

Relative population performance was first introduced in Balduzzi et al. (2019) as the performance of individual agents is meaningless in purely cyclical games. Intuitively, a relative population performance measure of $v(\mathfrak{B}, \mathfrak{D})$ implies that there exists a mixture policy in the population $\mathfrak{B}$ that achieves a payoff at least $v(\mathfrak{B}, \mathfrak{D})$ against all mixture policies in the opponent population $\mathfrak{D}$. This measure is defined formally in Definition A.1.

**Definition A.1** (Relative Population Performance). Given two populations of policies $\mathfrak{B}, \mathfrak{D}$ and let $(\mathbf{p}, \mathbf{q})$ be a Nash equilibrium over the zero-sum game on $\mathcal{U}_{\mathfrak{B}, \mathfrak{D}} \in \mathbb{R}^{M \times N}$, Relative Population Performance measures their relative performance: $v(\mathfrak{B}, \mathfrak{D}) := \mathbf{p}^{\mathbf{T}} \cdot \mathcal{U}_{\mathfrak{B}, \mathfrak{D}} \cdot \mathbf{q}$.

## A.3  NEUPL WITH STATIC INTERACTION GRAPH AND ABR BY RL

In this section, we describe a specialised implementation of NeuPL given *static* meta-graph solvers $\mathcal{F}$ with an approximate best-response operator implemented via deep Reinforcement Learning. The algorithm is described in Algorithm 4.

---

**Algorithm 4** Neural Population Learning by RL (static $\mathcal{F}$)

---

1: **function** RUN-EPISODES($\Sigma, \Pi_\theta, M$)                                    ▷ Visualized in Figure 10 (Right).
2:     $\mathcal{D}_\Pi \leftarrow \{\}; \mathcal{D}_Q \leftarrow \{\}$                        ▷ Initialize actor, critic trajectory buffers.
3:     **for** Episode $\in \{1, \ldots, M\}$ **do**                                ▷ Generate trajectories for $M$ episodes.
4:         Sample $\sigma_i \sim$ UNIQUE($\Sigma$) uniformly (excluding sink nodes) and $\sigma_j \sim P(\sigma_i)$.        ▷ Match-making.
5:         Sample trajectories $\mathcal{T}^{\sigma_i}, \mathcal{T}^{\sigma_j}$ from interactions between $(\Pi_\theta(\cdot|o, \sigma_i), \Pi_\theta(\cdot|o', \sigma_j))$.
6:         $\mathcal{D}_\Pi \leftarrow \mathcal{D}_\Pi \cup \mathcal{T}^{\sigma_i}; \mathcal{D}_Q \leftarrow \mathcal{D}_Q \cup \mathcal{T}^{\sigma_i} \cup \mathcal{T}^{\sigma_j}$.
7:     **return** $\mathcal{D}_\Pi, \mathcal{D}_Q$
8:
9: **function** NEUPL-RL-STATIC($\Sigma, T$)                                ▷ $\Sigma \in \mathbb{R}^{N \times N}$ the static interaction graph.
10:     Initialize $\Pi_\theta(a|o, \sigma_i)$, the task-conditioned policy.
11:     Initialize $Q_\theta(o, a, \sigma_i, \sigma_j)$, the action-value function.
12:     **for** Iteration $t \in \{1, \ldots, T\}$ **do**                                ▷ $T$ the number of iterations to run for.
13:         $\mathcal{D}_\Pi, \mathcal{D}_Q \leftarrow$ RUN-EPISODES($\Sigma, \Pi_\theta, M$)
14:         Optimize policy $\Pi_\theta$ from $\mathcal{D}_\Pi$ and $Q_\theta$ from $\mathcal{D}_Q$ by RL.

---

## A.4  NEUPL WITH ADAPTIVE META-GRAPH SOLVERS AND ABR BY RL

Building on Algorithm 4, we now extend to the case of *adaptive* meta-graph solvers $\mathcal{F}$ that are functions of empirical payoff matrices $\mathcal{U}$. Specifically, the set of meta-game strategies the set of policies seek to best-respond to are given by $\Sigma \leftarrow \mathcal{F}(\mathcal{U})$. This algorithm is described in Algorithm 5.

# B  EXPERIMENTS

## B.1  RUNNING-WITH-SCISSORS ENVIRONMENT

Figure 9 shows an example view of the *running-with-scissors* environment at the start of an episode. The dashed squares shows the type of resources initialized in the enclosed area with some of them consistently initialized with one type of resources while others randomly initialized with one of the

---

**Algorithm 5** Neural Population Learning by RL (adaptive $\mathcal{F}$)

1: **function** NEUPL-RL-ADAPTIVE($\mathcal{F}, K, T, M, \epsilon$)                                          ▷ $\mathcal{F}: \mathbb{R}^{N \times N} \to \mathbb{R}^{N \times N}$.
2:    Initialize $\Pi_\theta(a|o_\le, \sigma_i)$, the neural population network.
3:    Initialize $Q_\theta(o_\le, a, \sigma_i, \sigma_j)$, the action-value function.
4:    Initialize $\phi_\omega(\sigma_i, \sigma_j)$, the empirical payoff estimator.
5:    Let $\Sigma_c \in \mathbb{R}^{N \times N} \leftarrow \mathbf{1/N}$.                                          ▷ All-to-all interactions.
6:    **for** Epoch $\in \{1, \ldots, K\}$ **do**                                          ▷ $K$ epochs to run.
7:        $\forall i, j: \mathcal{U}_{ij} \leftarrow \phi_\omega(\sigma_i, \sigma_j)$ the payoff matrix over $N$ policies.      ▷ Re-compute payoff estimates.
8:        $\Sigma \in \mathbb{R}^{N \times N} \leftarrow \mathcal{F}(\mathcal{U})$ the interaction graph over $N$ policies.     ▷ Update the interaction graph.
9:        **for** Iteration $t \in \{1, \ldots, T\}$ **do**                                          ▷ $T$ iterations per epoch.
10:           $\mathcal{D}_\Pi, \mathcal{D}_Q \leftarrow$ RUN-EPISODES($\Sigma, \pi_\theta, M \times (1 - \epsilon)$)         ▷ $M$ episodes per iteration.
11:           $\_, \mathcal{D}_Q^\epsilon \leftarrow$ RUN-EPISODES($\Sigma_c, \pi_\theta, M \times \epsilon$)         ▷ $\epsilon$ the prop. of evaluation episodes.
12:           Optimize policy $\pi_\theta$ from $\mathcal{D}_\Pi$ and $Q_\theta$ from $\mathcal{D}_Q \cup \mathcal{D}_Q^\epsilon$ by RL.
13:           Optimize $\phi_\omega$ from $\mathcal{D}_Q \cup \mathcal{D}_Q^\epsilon$ by minimizing $\mathcal{L}_{ij}$.

---

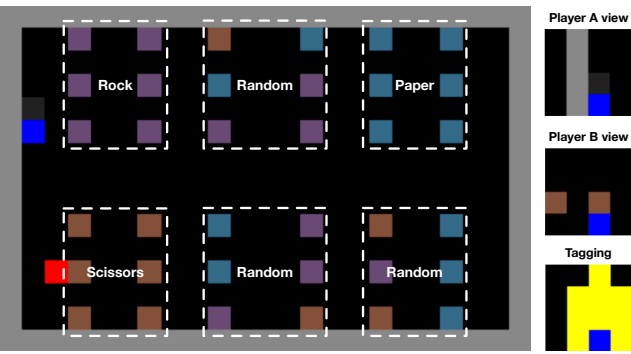

Figure 9: An example view of the *running-with-scissors* environment upon initialization.

three types. On the right, we visualize the 4x4 pixel observations of the two players. The visual observation is oriented along each player's forward direction. In addition to the visual observations, each player observes their current inventory of the three types of resources, expressed in terms of their normalized ratios. Each player is initialized with an equal weight inventory at the start of an episode. To move around, a player can turn left, turn right, strafe left, strafe right, move forward or move backwards. Finally, each player can proactively seek out its opponent and "tag" it to terminate the game. A player is considered tagged if it falls into the tagging area of its opponent. Bottom right view illustrates the shape of the tagging area in front of a player. If neither player tags its opponent, the game ends after a fixed number of 500 steps. On the terminal step, the game resolves by comparing the inventory of both players and the rewards are assigned according to the classical anti-symmetric payoff matrix of *rock-paper-scissors*. On all other steps, both players receive zero rewards.

### B.2    CONDITIONAL NETWORK ARCHITECTURE FOR NEUPL

Figure 10 (Left) illustrates the general network architecture of a NeuPL population for a typical Q-learning based RL agent. In particular, the encoder, memory and policy head network modules are shared across all policies within the neural population with the conditioning variable $\sigma$ introduced at the final policy head layer via concatenation. This reflects the hypothetical Game-of-Skills geometry proposed in Czarnecki et al. (2020), where each policy can be understood as a point within a "spinning-top" volume. Each policy is interpreted as a combination of strategy-agnostic transitive skills (e.g. movement skills for embodied agents; representational capability of past observations in partially-observed games) and a cyclic strategic element that decides which mixture policy to best-respond to, leveraging its transitive skills.

At a high-level, the goal of NeuPL is to represent a compact set of policies that corresponds to the top layer of this "spinning-top" geometry, or the NE of the game. As such, our proposed conditional network architecture facilitates the sharing of strategy-agnostic transitive skills across policies, while isolating strategy-specific decisions that may call for different actions in the same state within the

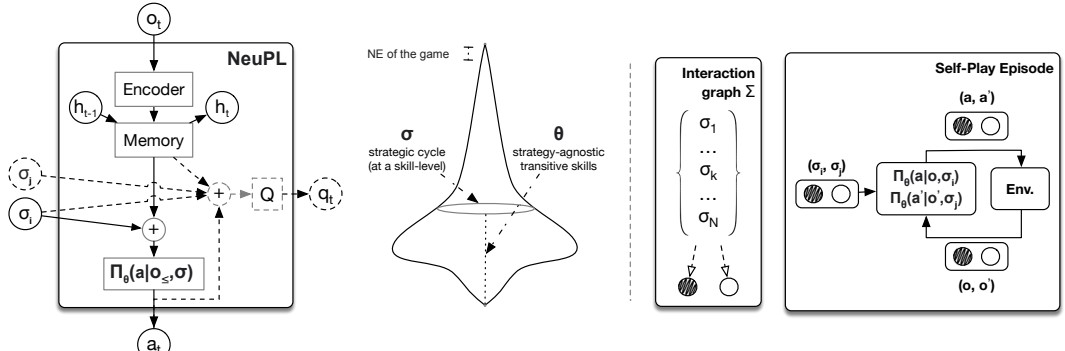

Figure 10: **(Left)** The conditional neural population network architecture for a Q-learning based RL agent. The "$\bigoplus$" sign denotes a concatenation of all inputs tensors and dashed components are only used for training (and not necessary for acting); **(Middle)** A diagram illustrating the separation between the strategy-agnostic transitive skill dimension captured by the shared parameters $\theta$ and the strategic cycles captured by $\sigma$ at a skill-level; **(Right)** A diagram illustrating the mechanism of a NeuPL self-play episode: at the start of an episode, a pair of conditioning vectors are sampled $(\sigma_i, \sigma_j)$ from the interaction graph $\Sigma$. The pair of policies, using the same conditional policy network $\Pi_\theta(a|o_\le, \sigma)$, act simultaneously in the environment. The circles correspond to the conditioning vectors, observations and actions for the exploiter policy (shaded) and its opponent (blank).

final opponent-conditioned policy head network, mitigating the concern of negative transfer. We note that while we limited ourselves to the simplest conditioning architecture in this work, investigating conditional network architecture with suitable inductive biases could be an interesting future direction.

### B.3 Hyper-Parameters & Training Infrastructure

For experiments on the normal form game of *rock-paper-scissors*, the "Encoder" and "Memory" components are omitted and the policy head network $\pi_\theta$ alone is parameterized and learned, with its only input $g$. The policy and action-value networks are both parameterized by 4-layer MLPs, with 32 neurons at each layer. We use a small entropy cost of $0.01$, learning rates of $0.001$ and $0.01$ for the main networks and the MPO dual variables (Abdolmaleki et al., 2018) respectively. As is typical in an MPO agent, the online network parameters are copied to target networks periodically at every 10 gradient steps. For the spatiotemporal, partially observable game of *running-with-scissors*, we use a small convolutional network to encode the agent's partial observation of the environment (a 4x4 grid surrounding itself). The encoding is further concatenated with the agent's own inventory information and encoded through a 2-layer MLP with 256 neurons each, with `relu` activation. The memory module corresponds to a recurrent LSTM network, with a hidden size of 256. The policy, action-value networks are parameterized as 4-layer MLP networks. The learned payoff estimator $\phi_\omega(\sigma_i, \sigma_j)$ is parameterized as a 3-layer MLP network. The learning rate of the agent networks is set to $0.0001$ while the MPO dual variables are optimized with a learning rate of $0.001$. The online network parameters are copied to target networks every 100 gradient steps. For the MuJoCo Football environment, we used a domain specific encoder network that encodes egocentric observations of each player individually. The state representation is then implemented as a weighted sum of per-player embedding, using a learned attention mask. Instead of outputting a discrete categorical action profile in each state, the policy head network is trained to output a Gaussian distribution with learned mean and variance distribution parameters. As is common in continuous control literature, we used `elu` as the activation function between network components. The rest of the network architecture and hyper-parameters remain consistent with that of *running-with-scissors*. The same network architecture and hyper-parameters are used across all experiments for the same environment.

Figure 10 (Right) illustrates how self-play experience data is generated for a NeuPL population at training time. At the start of an episode, a pair of conditioning vectors is sampled $(\sigma_i, \sigma_j)$ and used to condition the pair of policies interacting in the game. In *running-with-scissors*, each NeuPL experiment uses 128 actor workers running the policy environment interaction loops and a single TPU-v2 chip running gradient updates to the agent networks. The same computational resources are

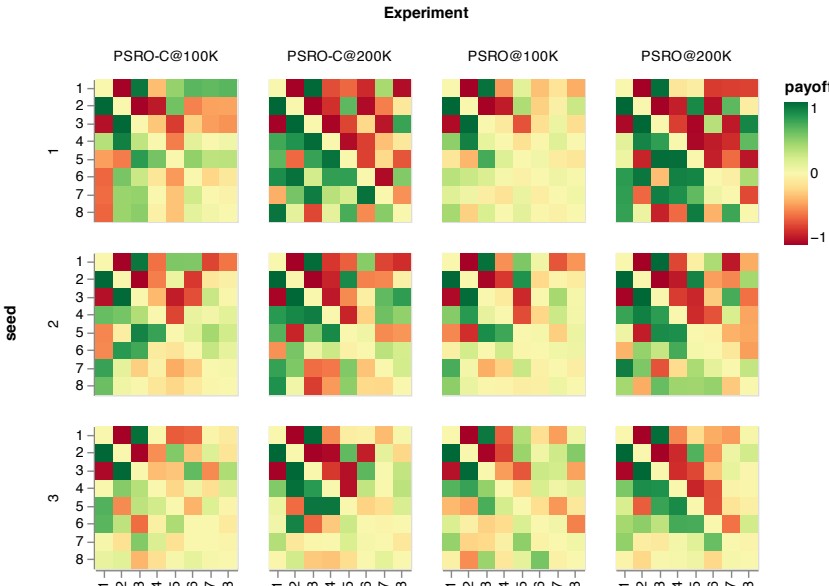

Figure 11: Visualization of empirical payoff matrices for each PSRO population after 8 iterations across three independent trials. Each iteration lasts 100k or 200k gradient steps. PSRO-C indicates that the policy trained at iteration $i + 1$ is initialized from the policy obtained at iteration $i$ instead of from random initialization. A payoff of 1.0 (-1.0) indicates a win (loss) probability of 100% for the row player.

used across different maximum population size for NeuPL as well as the PSRO baseline experiments. For MuJoCo Football, 256 CPU actors are used per learner. For the game of *rock-paper-scissors*, a single CPU worker is used instead. In *running-with-scissors*, all experiments converge within 2 days wall-clock time. For all experiments using NeuPL, an evaluation split $\epsilon = 0.3$ is used.

## B.4 BASELINE PSRO EXPERIMENTS

Figure 11 visualizes the empirical payoff matrices obtained in 12 independent experiments. In each experiment, we develop a discrete population of policies via 8 iterations of PSRO training. In particular, we show the payoff matrices between trained policies where each PSRO iteration lasts 100k or 200k gradient steps. PSRO-C further shows the effect of initializing each iteration by the policy obtained at the end of the preceding iteration instead of random initialization as prescribed in (Lanctot et al., 2017). We note that across all experiments, policy populations successfully recover the *rock-paper-scissors* dynamics among the first three policies. In most but not all trails, PSRO also manages to learn a reliable exploiter of the mixture of the three pure strategies. Finally, we note that continued training appears to facilitate the discovery of richer strategic cycles, and so does allowing each PSRO iteration to perform more gradient updates.

## B.5 SENSITIVITY TO THE CHOICE OF HYPER-PARAMETERS

Figure 12 investigates the sensitivity of NeuPL to the choice of hyper-parameters. In particular, NeuPL introduces two additional hyper-parameters: the proportion of simulation episodes used for learning pairwise empirical payoffs ($\sigma$ in Algorithm 5) and the interval between interaction graph update ($T$ in Algorithm 5). Intuitively, if $\epsilon$ is too low, the empirical payoff estimator risk being under-trained when its estimations are used by the meta-graph solver while a value too high would delay the policy learning due to insufficient simulation data being generated for policy learning. When it comes to the meta-graph update interval, a value too high may slow down strategic exploration if the approximate best-response operators have already converged to best-responses while a value too low may lead to noisy gradients in the optimization process, as the set of learning objectives may change too frequently.

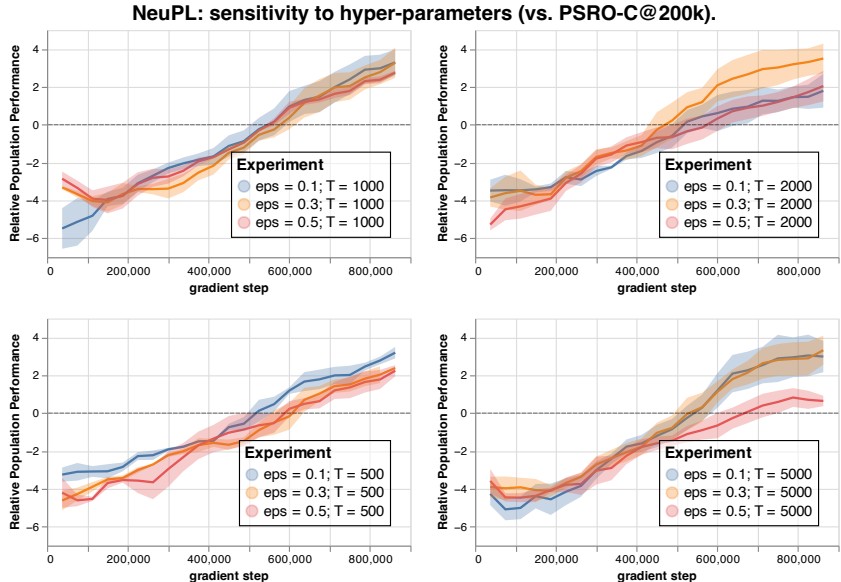

Figure 12: Hyper-parameter sweep across proportion of matches used for empirical payoff evaluation ($\epsilon$) and interaction graph update interval in gradient steps ($T$). Population-level performance is measured in RPP, relative to the same held out population `PSRO-C@200k` with `seed = 1`.

In practice, we observe that NeuPL is reasonably robust to these hyper-parameters across a wide range of choices. Across all experiments, we evaluate the NeuPL population's relative population performance against a fixed, held-out population obtained via `PSRO-C@200k, seed = 1`.

### B.6 Test-time Execution of Learned Policies

A distinctive property of population learning is that we obtain a population of policies that can be executed individually at test time. In NeuPL, the population of policies is jointly defined by two elements: a conditional network $\Pi_\theta(\cdot | o_{\leq t}, \sigma)$ and a set of meta-game strategies $\Sigma = \{\sigma_i\}_{i=1}^N$, derived from the empirical payoffs between policies. To execute a mixture policy defined over the population, it suffice to execute $\Pi_\theta(\cdot | o_{\leq t}, \hat{\sigma})$ with $\hat{\sigma} \sim Pr(\Sigma)$.

Specific mixture policies have well-understood properties. For instance, playing the NE mixture policy implies that an opponent who has access to the same set of policies is indifferent to playing any mixtures. This could be a principled option in the absence of prior knowledge about one's opponent.

## C Convergence Proofs

In this section we detail the conditions upon which NeuPL will converge to a unique sets of policies. First, we discuss the theory when the interaction graph is static and grounded (Section C.1). Then we discuss the theory when the interaction graph is a function of the payoff matrix, a so-called meta-graph solver (Section C.2). Finally we discuss a specific function, Nash Equilibrium meta-graph solver, that is popular in the literature, and prove NeuPL's convergence to a normal-form Nash Equilibrium under certain conditions (Section C.3).

### C.1 Grounded Interaction Graphs

**Assumption C.1** (Unique, Exact, Finite Best Response). *Assume that we have access to a best response (BR) operator that responds to a policy, $\pi_n$, (or equivalently a mixture over policies) exactly converges to a unique best response, $\pi_{n+1} = BR(\pi_n)$, in finite time.*

When using RL as the BR operator, uniqueness can be approximated with a small entropy term on policy being optimized. Therefore, in the situation where there is more than one best response, the

BR operator will opt for the one with maximum entropy. This is a common additional loss that is often added to RL agents, and is also believed to aid exploration (Haarnoja et al., 2018).

The second assumption, that the BR will be exact, is idealised. With sufficient model capacity, enough time, and appropriate hyper-parameter annealing schedules, RL can get close to an exact BR.

The final assumption, that the BR operator converges in finite time, is assumed because proofs follow easily from this assumption. It may be possible to relax this assumption to converge as $t \to \infty$, but we leave this for future work.

**Definition C.1** (Grounded Interaction Graph). An interaction graph's edges define how policies should best respond to each other. We call an interaction graph *grounded* if its structure imposes convergence to a unique set of policies, where each policy is a unique, exact, finite BR over opponent policies defined by the interaction graph.

**Theorem 1** (Grounded Lower Triangular). *All lower-triangular interaction graphs ($\Sigma_{i \geq j} = 0$) are grounded.*

*Proof.* The first row of a lower triangular matrix is all zeros meaning that the first policy does not respond to any other policies. The first policy is therefore static (and arbitrary) and does not change over time. The second row only responds to the first policy. Since the first policy is static and we are using a unique, exact and finite BR, the second policy will converge exactly to a unique policy. The $n$th row will respond to the $n-1$ previous policies. Since over time all previous policies will converge to uniquely, so will the $n$th policy. $\square$

It is easy to imagine other grounded interaction graphs with interesting structure, however we will only focus on lower-triangular graphs for the purpose of this section. Research into other grounded interaction graphs may be interesting future work.

**Definition C.2** (Static Interaction Graph). We call an interaction graph *static* if it does not change over time: $\Sigma^t = \Sigma$.

We now prove our first NeuPL result, that it can find an exact $N$-step best response under certain assumptions.

**Theorem 2** (NeuPL Static Lower-Triangular Exact $N$-Step Best Response). *NeuPL with a static, lower triangular interaction graph, an arbitrary fixed initial policy, $\pi_0$, and $N$ more policies, will converge to an exact $N$-step best response, assuming that the BR is unique, exact, and converges in finite time.*

*Proof.* The bulk of this proof can recycle the arguments in Theorem 1. Note that NeuPL trains its conditional policies in parallel according to an interaction graph. If that graph is lower-triangular, it is grounded, and NeuPL will find the conditioned policies corresponding to the $N$-step best response. $\square$

## C.2 Grounded Meta-Graph Solver

It is possible to extend this result on any deterministic function acting on a sub-payoff, producing a lower triangular interaction graph.

**Definition C.3** (Grounded Meta-Graph Solver). A meta-graph solver is a function that takes the payoff matrix as an argument and outputs an interaction graph as an output. We call a meta-graph solver, $\mathcal{F}$, *grounded* if, when using unique, exact, finite BRs, it converges to a grounded interaction graph in finite time.

**Theorem 3** (Any Deterministic Lower-Triangular Meta-Graph Solver is Grounded). *Any deterministic function, $\mathcal{F}$, that maps a* sub-payoff, $\mathcal{U}_{<i,<i}$, *to a row, $\Sigma_{i,<i}$, in a lower-triangular interaction graph is a grounded meta-graph solver.*

$$\Sigma_{i,<i} = \mathcal{F}(\mathcal{U}_{<i,<i}) \tag{2}$$

*Proof.* The difficulty here is that in general, as the policies change, so will the payoff, and hence so will interaction graph, etc. Similar to the arguments in Theorem 1 the first policy is static and does not

change. The second policy can only respond to the first policy (under the lower-triangular constraint), so the meta-graph solver has no flexibility to do otherwise, and therefore the second policy will also converge to a unique result. Note that as these policies converge, so will their sub-payoffs, $\mathcal{U}_{<3,<3}$. Therefore, for row $n$ any deterministic mapping $\mathcal{F} : \mathcal{U}_{<i,<i} \rightarrow \Sigma_{i,<i}$ will result in a unique set of policies. □

We can now make a further claim about NeuPL: that it will converge to a unique set of conditioned policies under certain grounded meta-graph solvers.

**Theorem 4** (NeuPL Deterministic Lower-Triangular Exact $N$-Step Best Response). *Assuming any deterministic lower-triangular meta-graph solver, an arbitrary fixed initial policy, $\pi_0$, and $N$ more policies, NeuPL will converge to an exact $N$-step best response, assuming a unique, exact and finite BR.*

*Proof.* Similar in structure to Theorem 2 and Theorem 3. □

Of course, the more general result, that NeuPL will also converge for any grounded meta-graph solver is also true.

### C.3 Nash Equilibrium Meta-Graph Solver

Note that the maximum entropy Nash Equilibrium (MENE) is such as grounded meta-graph solver. This will result in an algorithm similar to PSRO-Nash (Lanctot et al., 2017) or Double Oracle (McMahan et al., 2003).

**Theorem 5** (NeuPL Nash Exact $N$-Step Best Response). *Using an interaction-graph function that maps payoff to a lower triangular NE distribution:*

$$\mathcal{G}_{n,<n} = NE(U_{<n,<n}) \tag{3}$$

*With a such a function, an arbitrary fixed initial policy, and sufficiently large $N$ policies, NeuPL will converge to an exact normal-form Nash Equilibrium (NE), assuming a unique, exact, finite BR.*

*Proof.* Starting from the proof in Theorem 4, we follow additional arguments from DO (McMahan et al., 2003) to prove that for sufficient $N$ we will converge to a normal form NE. □

Of course, many other algorithms can be recovered using specific meta-graph solvers (Muller et al., 2020; Marris et al., 2021).

## D   Generality of NeuPL and Discussion on Future Works

For simplicity of presentation, we focused on the simple setting of learning from scratch in symmetric zero-sum games. In this section, we discuss elements needed towards applying NeuPL more generally from several aspects.

### D.1   Incorporating Prior Knowledge in NeuPL

As we alluded to in Section 1.2, the formulation of NeuPL offers a principled way to incorporate prior knowledge in the form of pre-trained policies. In short, pre-trained policies can be incorporated in the same way as the *sink* policy, with the requirement that it can only gain in-edges in the interaction graph. Figure 13 offers an illustration of such an example, where the population includes 2 pre-trained policies ($\Pi_\theta(\cdot|o_{\leq t}, \sigma_1)$ and $\Pi_\theta(\cdot|o_{\leq t}, \sigma_2)$) while $\Pi_\theta(\cdot|o_{\leq t}, \sigma_3)$ is optimized to best-respond to a mixture over its predecessors according to a suitable meta-graph solver. As an implementation details due to the use of neural networks, $\sigma_1$ and $\sigma_2$ need to be unique vectors and they need to be treated specifically in the corresponding MGS as they no longer represent valid probability distribution and should be excluded from policy training, similar to the treatment of the *sink* policy.

Finally, we note that our proposed conditional network architecture is in fact synergistic with the use of pre-trained policies, beyond naively including pre-trained expert policies as opponents in

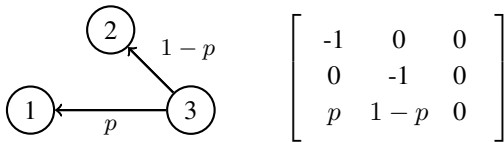

Figure 13: Example NeuPL experiment with an interaction graph incorporating pre-trained policies.

the population. This is because the shared action-value function $Q_\theta(o_{\leq t}, \sigma_i, \sigma_j)$ would learn to predict expected returns between fixed pre-trained expert policies, kick-starting the learning of encoder and memory components in the underlying game. As proposed in Figure 10, this kick-started representation learning process should in principle transfer to the learning of other policies within the neural population as well, rendering it an attractive proposal.

### D.2 GENERALIZATION TO N-PLAYER GENERAL-SUM GAMES

NeuPL can in principle extend to n-player general-sum games. We offer elements of the solution in this section while deferring thorough investigation in this direction to dedicated future work.

Consider an $n$-player general-sum game where the $i$-th player can play one of $M^i$ policies, the induced meta-game corresponds to a normal-form game between $n$ players, each selecting a policy to execute for an episode for one player. This yields $n$ empirical payoff tensors $\mathbf{U} = \{\mathcal{U}^i\}_{i=1}^n$ with $\mathcal{U}^i \in \mathbb{R}^{M^1 \times M^2 \cdots \times M^n}$ the payoff tensor for player $i$, given all players' policy selections.

**Coarse Correlated Equilibrium (CCE):** in this setting, solving for NE becomes intractable but solution concepts such as CCE could be readily used instead (Marris et al., 2021). This leads to a solver that takes on the form $\mathcal{P} \leftarrow \mathcal{F}(\mathbf{U})$ with $\mathcal{P} \in \mathbb{R}^{M^1 \times M^2 \cdots \times M^n}$ a joint-distribution over the cartesian product of policy choices across $n$ players. This solver can thus be used in place of the meta-strategy solver, similarly and repeatedly invoked by the meta-graph solver. Note that instead of obtaining a marginal distribution for a given player as in NE, CCE offers the joint distribution which can be marginalized for each player.

**Heterogeneous Neural Populations:** in the n-player setting, different players may take on different roles in the game and admit entirely different observation, action spaces. This implies that heterogeneous neural populations are needed. Specifically, each player's policies can be represented by its own neural population of maximum capacity $M^i$ and its own $n$-dimensional marginal interaction *tensor*. The $i$-th neural population is defined by a conditional model $\Pi_\theta^i(\cdot|o_{\leq t}, \sigma^i)$ and a marginalized interaction tensor $\Sigma^i \in \mathbb{R}^{M^1 \times M^2 \cdots \times M^n} = \{\sigma_k^i\}_{k=1}^{M^i}$, derived from incremental CCE joint distributions.

