# OpenReview forum: "NeuPL: Neural Population Learning"
_ICLR.cc/2022/Conference — ICLR 2022 Poster_

### Official Review · Reviewer_TfQw · 2021-10-24

**Correctness:** 3
**Technical Novelty And Significance:** 3
**Empirical Novelty And Significance:** 3
**Recommendation:** 8
**Confidence:** 4

**Main Review:**

**Review**
This paper offers two ideas (1) concurrent training of new responses and fine-tuning of previous responses through paramterizing a single policy with the other-agent strategies, and (2) a method for representing population based training algortihms through interaction graphs. The former seems a reasonble way to prevent unnecessary relearning; however, it raises concerns of policy collapse, negative transfer, and questions of what exactly is transferred. Despite concerns, it appears that empirically this a direction worth exploring. The later, is a new lens to view population based training algorithms from, and I believe offers the community a language to discuss algorithms with that has room for fruitful discoveries.

**Major Comments**

 - My impression from reading this paper is that the authors are proposing NeuPL as a more "general class of population learning algorithms" than PSRO (Sec 1.2). However, this seems very disingenuous because where NeuPL offers a few nicer generalizations it also restricts the class of games that can be studied. The authors need to fairly discuss their work limitations wrt prior work. In particular, acknowledging explicitly NeuPL is restricted to zero-sum symmetric games with fixed player sizes (as the policy's observation space is dependent on the game's number of players), where PSRO is not. Another area for limitations is policy collapse, or gradient interference when co-training dramatically different or contrary responses.
 - The literature review is missing related work and fails to place this work in the context of prior work that is shallowly cited. Last year's ICLR had a work that [1, 2] looked at transferring response knowledge across PSRO epochs, directly relevant to the problems being addressed here. Moreover, P2SRO [3] is shallowly cited when it is co-training across PSRO epochs. I also think it would be advantageous to compare and contrast with Self-Play, directly discussing that the changes are an expanded observation space alongside the matchmaking. Otherwise, using self-play to refer to playing against homogenously parameterized policies must be inferred. This work also has some indirect connections to [4], maybe not worth mentioning explicitly (opponent strategies used at training time instead of inference).
 - Comparing methods using only RPP, a dynamic measurement, may not paint a detailed enough picture to truly understand relative performance of two methods. For example, assuming weak RL, one method may settle into Rock-like policies while another into Paper-like policies. Creating a strict ordering in performance, and then chase each-other around the search space simply due to a staggering of learning. It would be beneficial to include absolute measurements relative to known solutions, or compare relative to a held-out static rich pool of evaluation policies. In other words, the algorithms have no equilibrium selection bias.
 - A major claim that the authors seek to address is that good-responses are being generated instead of best-responses. Have the authors considered that this may in fact be a feature? Empirically, I have noticed that noise introduced can be beneficial for strategy space exploration. Serendipitously, I could not find a direct argument that NeuPL addresses this problem, could the authros please elaborate? I suspect that the continuous training of all policies is the argument, but it's not clear why longer training time actually solves this problem. In my experience the truncation is because the reward marginals for exponentially more training are tiny.
 - I am having trouble reconstructing PSRO from the variation of PSRO presented in Algorithm 2. I suspect this is because "N", "M", and "N-step ABR" are undefined or inconsistently used. M is not ever defined, which I believe is being used here to track PSRO iterations, but no outer loop for PSRO is presented. Is N-step BR meant to refer to training a new policy for each i \in N player? If so, for symmetric games, as this work considers, only a single BR step is necessary (train i vs -i, and give all players the new policy).
 - How were the hyperparemeters selected for NeuPL and the baseline methods? Exploration schedules in particular seem particularly challenging to define well for NeuPL.


**Minor Comments (not impacting review)**

- Sec 0, Par 3, "... network via self-play" I think overloading self-play this early on could be very misleading given its established definition.
- Sec 1.2, Par 2, "opponent's meta-game strategy \sigma_i" I think you want \sigma_{-i}
- A major limitation of this work is that policies must be homogenous. So for example, if you wished to initialize the population with a diverse set of known heuristic strategies this would require some changes and not offer the same benefits that could be gleamed from competiing methods. Not that this is a setting realized currently in the literature.
 - Truncation of training still seems necessary for Alg-5, where it in effect represents a trade-off for good-ness of response and cost incurred from operations on the interaction graph. Did the authors explore looking at this dimension at all?

[1] Smith, Anthony, Wellman. Iterative Empirical Game Solving ia Single Policy Best Response. 2021.
[2] Smith, Anthony, Wellman. Learning to Play against Any Mixture of Opponents. 2020.
[3] McAleer, Lanier, Fox, Baldi. Pipeline PSRO: A Scalable Approach For Finding Approximate Nash Equilibria in Large Games.
[4] Foerster, et al.. Stabilising Experience Replay for Deep Multi-Agent Reinforcement Learning. 2017.

**Summary Of The Paper:**

Population based training (PBT) algorithms progressively grow a set of policies by adding best-responses to mixtures of the existing population. When RL is used as a best-response method the new policy is generally not a best-response but instead a good response. Moreover, the good-response is initialized tabula rosa potentially relearning responses to policies responded to in previous iterations. To address these concerns, this work proposes NeuPL, where a single policy is trained to represent the entire population by conditioning it on the opponent's mixture. Moreover, this proposed algorithm introduces the use of interaction graphs as a means to codify match-making in a continuously training population. NeuPL is restricted to zero-sum symmetric games with fixed player sizes.

**Summary Of The Review:**

Overall I think the paper is sensible to appear at ICLR, but would not fight for it or be upset if it was not accepted.

The ideas presented within the work are interesting and provide a way to frame PBT methods and could facilitate the design of many future algorithms. These ideas are supported with some empirical demonstrations; however, my concerns regarding: insufficient discussion of place in literature, absent discussion of how hyperparameters are arrived at, and lack of quantitative analysis on the later experiments; leave me with reservations.

---

> ### Author Response · Authors · 2021-11-19
> **Updated comments to reviewer TfQw**
>
> We thank the reviewer for recognising the importance of our work and for raising valid concerns regarding our initial submission. We have addressed most of these issues and updated our manuscript accordingly. We now address specific questions as follows.
>
> **R4/Q1: the parameter sharing raises concerns of negative transfer, and questions of what exactly is transferred.**
>
> This is an important question and related to `R1/Q1`. What we hope to be transferred across policies are _strategy-agnostic skills that support the implementation of diverse strategies_ (as defined in [1]). Our results (Figure 6) verify this hypothesis, showing that pre-trained, strategy-agnostic representation indeed transfers to learning exploiters to previously unseen, sophisticated opponents.
>
> Consider the experiment shown in Figure 4: early on, $\\{ \Pi_\theta(\cdot|s, \sigma_i) \\}^3_{i=2}$ each optimises against a “pure-resource” opponent. At this stage, past observations of the opponent is *irrelevant* and memory is only used for navigation; $\Pi_\theta(\cdot|s, \sigma_4)$ then faces a mixture of pure-resource opponents: the adaptive policy must now categorise its opponent as one of the three pure-resource opponents, based on past observations. This categorisation would then fall short, once the opponent learns to “pretend to collect rock”, as implemented by $\Pi_\theta(\cdot|s, \sigma_5)$. We believe that such nuanced, strategy-agnostic representation skill is transferable across strategies and is the main contributor to the positive transfer in Figure 6.
>
> The concern of negative transfer is shared across multi-task learning [2] which motivated our network design where the representation layers are strategy-agnostic. If two strategies call for different actions in the same state, such decision pathway would be narrowly encapsulated within the policy head whereas the strategy-agnostic representation layer remains relevant across strategies. As we used the simplest conditional network architecture in this work, we believe exploring alternative architectures could lead to productive future works.
>
> **R4/Q2: the authors should discuss the limitations of NeuPL and fairly discuss its limitations with respect to PSRO.**
>
> We apologise that our initial submission may have given the wrong impression that we claimed NeuPL to be more general than PSRO in all aspects. We have updated our writing to be more precise in this regard.
>
> **generality compared to PSRO:** our intention is to point out that unlike PSRO, where best-responses are learned in an iterative, lower-triangular, grounded way ($\pi_i$ cannot learn to best-respond to $\pi_j$ with $i < j$), NeuPL allows the specification of any, possibly ungrounded or cyclic interaction graphs. One such example is the learning of a strategic cycle in RPS as we have shown in Figure 2 (middle).
>
> **symmetric-zero sum:** we focused on the symmetric zero-sum setting in this work and we have updated our writing to make this clearer. Nevertheless, we note that the main idea behind NeuPL can be feasibly extended to n-player general-sum games. We have now described elements of the solution in Appendix D. We acknowledge that dedicated investigations are needed in this direction.
>
> **fixed player size:** we agree with the reviewer that the fixed maximum population size remains a practical limitation due to the use of neural networks. However, we have shown in Figure 5 that the maximum population size does not meaningfully affect the rate of convergence in practice. The number of distinct policies represented by the neural population evolves at similar rates, across experiments with different maximum population sizes. In all cases, the number of unique policies converged to around 12. This shows that the maximum capacity specified has limited impact on learning and can be "over-specified" at the costs of over-parameterisation.
>
> **R4/Q4: the authors should contrast NeuPL to prior works that similarly seek to scale up population learning methods such as PSRO.**
>
> We thank the reviewer for bringing these relevant prior works to our attention. We have updated our writing to include specific comparisons between NeuPL and these prior works. A more thorough discussion is provided in our answers to R3/Q1.
>
> **R4/Q5: the term self-play is misused and needs clarifying.**
>
> We agree with the reviewer that our use of “self-play” is overloaded and confusing. Our intention was to highlight that NeuPL enables game-theoretic population learning while requiring the computation infrastructure identical to that of self-play. We have now updated our writing accordingly.
>
>
> [1] Czarnecki, Wojciech M., Gauthier Gidel, Brendan Tracey, Karl Tuyls, Shayegan Omidshafiei, David Balduzzi, and Max Jaderberg. "Real World Games Look Like Spinning Tops." Advances in Neural Information Processing Systems 33 (2020).
>
> [2] Caruana, R. (1997). Multitask learning. Machine learning, 28(1), 41-75.
>
> [... continuing ...]

---

> > ### Author Response · Authors · 2021-11-19
> > **continuing**
> >
> > **R4/Q6: all experiments relied on RPP. Why was this metric used rather than evaluating against static held out populations?**
> >
> > In classical games, the quality of a population of policies can be measured by its exact exploitability. Unfortunately, such methods cannot scale to larger games. An alternative as suggested is to introduce a held-out population though it does not remove the need for us to justify the richness of such populations in the first place. For this reason, we presented all 12 independent baselines in Figure 11. This should offer evidence that the PSRO populations reasonably capture the strategic richness of the game. Most populations indeed represent the pure-resource policies, their adaptive counter-strategy and in turn, its deceptive counter.
> >
> > As a population-level metric, RPP is also robust to replicas (unlike e.g. Elo) and consistent improvement in RPP implies true strategic dominance. The pattern of consistently improving RPP that we observed in Figure 7 cannot be simply explained by weak RL and we presented a more likely explanation in Figure 6.
> >
> > **R4/Q7: noise introduced during strategy exploration by “good-responses” may even be beneficial for empirical strategy space exploration. Why is it framed as a drawback?**
> >
> > We agree with the reviewer that “noise” could have its empirical benefits for strategic exploration. Nevertheless, convergence proofs typically rely on exact best-response oracles. It is for this reason that we consider good-response populating the population undesirable: not because it would necessarily perform worse empirically but that it deviates from the assumptions made in known theoretical analysis.
> >
> > **R4/Q8: how are the hyper-parameters chosen? They seem difficult to tune.**
> >
> > All our experiments on the `running-with-scissors` domain used the same set of default MPO hyper-parameters without extensive search. NeuPL introduces two hyper-parameters: the evaluation split $\epsilon$, which is set to 30% across all experiments and the interaction graph update interval $T = 1000$ gradient steps.
> >
> > We have now run additional ablation studies to understand the sensitivity of NeuPL. Our results in Appendix B.5 showed that NeuPL is robust to $\sigma$ and $T$ across a range of values.
> >
> > **R4/Q9:A major limitation of this work is that policies must be homogenous and prior knowledge such as pre-trained policies cannot be easily incorporated.**
> >
> > This is an interesting question that we did not discuss in our initial submission. In fact, a different interpretation of our usage of a "pure-rock" initial policy is that we have indeed introduced prior knowledge to our policy population, namely that the initial policy is known to be exploitable and is thus a reasonable starting point for strategic exploration.
> >
> > We have introduced an additional section in Appendix D.1 that more precisely describes how multiple pre-trained policies can be incorporated in NeuPL in a principled and *synergistic* fashion. In particular, learning action-value functions for the pre-trained policies can help kick-start the representation learning for the entire neural population.
> >
> > **R4/Q10: Sec 1.2, Par 2, "opponent's meta-game strategy \sigma_i", it should be \sigma_{-i} instead**
> >
> > The $i$-th policy in the neural population is conditioned by $\sigma_i$ which corresponds to a mixture that *the said policy tries to best-respond to*. In the language of PSRO, such a mixture policy corresponds to an opponent's meta-game mixture strategy which is the object referred in this statement. As such, $\sigma_i$ should be used instead of $\sigma_{-i}$ in this context. Please let us know if this remains unclear as it is a common source of confusion.
> >
> > **R4/Q11: Truncation of training still seems necessary for Alg-5, where it in effect represents a trade-off for good-ness of response and cost incurred from operations on the interaction graph.**
> >
> > We understand the question as referring to L9 of Algorithm 5. At a high-level, we note that an update to the interaction graph need not imply a truncation in learning a best-response.
> >
> > Consider the case where NeuPL implements PSRO-Nash. Consider iteration i where policy $\Pi_\theta(\cdot|o_{\le t}, \sigma_i)$ seeks to best-respond to the NE over $\\{ \Pi_\theta(\cdot|o_{\le t}, \sigma_j) \\}_{j < i}$.
> >
> > If $\\{\Pi_\theta(\cdot | o_{\le t}, \sigma_j) \\}_{j < i}$ have each converged to be the best-responses to their respective mixture opponents, then $\sigma_i$ would converge to a fixed distribution, namely the NE over its predecessors.
> >
> > In practice, this implies that subsequent interaction graph update would make no change to $\sigma_{\le i}$, hence there would be no truncation to the best-response learning to the NE over $\\{\Pi_\theta(\cdot|o_{\le t}, \sigma_j)\\}_{j < i}$, until the experiment is stopped.
> >
> > Incidentally, this line of reasoning underlies our convergence proof, wherein we assume that the meta-graph solver is deterministic given payoff matrices.

---

> > > ### Comment · Reviewer_TfQw · 2021-11-19
> > > **Response to Rebuttal**
> > >
> > > I'd like the thank the authors for taking the time to provide nicely itemized responses to all of the reviewers' concerns. These have resolved most of my questions and thoughts regarding the details of this work. There are only two final comments that I would like to make that do not require a rebuttal.
> > >
> > > **R4/Q1**
> > > I agree that this experiment lends evidence towards the acceptance of the hypothesis; however, I don't think it's sufficient to fully verify the hypothesis. Unfortunately, I cannot think of a way to truly prove this with our modern understanding of MARL methods. I am not holding this against this work. A compromise would be to soften the tone to not suggest such as strong result.
> > >
> > > **R/Q2**
> > > I agree with the authors in that they need not have empirical baselines for the Q-Mixing algorithms and P2SRO. As they've noted, comparison across these methods is not intuitive and may lead to more misleading conclusions than helpful take-aways. As I originally wrote in my review, I do think that a serious discussion comparing these methods is absent from the original paper and is necessary. The revised version of the paper begins to work towards addressing this concern. I am a bit disappointed to see that the authors have chosen an "aggressive" approach to their literature review, as is so common in our field. I don't think it's important to attack prior work to justify current work. In particular, for the Mixed-Oracles comparison, it is true to observe that RL is never used to train against a Bayes-optimal objective; however, empirically this does not seem to inhibit their work. I think it would be more effective to discuss (1) how NeuPL affords the opportunity of transferring/reusing representations whereas Mixed-Oracles complexity grows in the size of the opponent support, and (2) how Mixed-Oracles likely offers faster learning of more exploitative best-responses (as it trains specialized policies; this could be where mentioning Bayes-optimal objectives is reasonable). I realize that you're at your page limit and cannot have as rigorous of a conversation as I would hope, but I urge the author's consider a more constructive and scientific approach to comparisons.

---

> > > > ### Author Response · Authors · 2021-11-19
> > > > **Minor clarification on our review of Mixed-Oracles**
> > > >
> > > > We thank the reviewer for providing timely feedback and we are glad to hear that our revision has made some progress towards positioning our work in the field.
> > > >
> > > > **R4/Q2**: we sought to be concise in highlighting the key differences between NeuPL and prior works. Unfortunately due to the character limit we had to take out a lot of context from our initial draft of the rebuttal (which still spanned two full comments!).
> > > >
> > > > In particular, our initial draft included a paragraph that explained how the idea of *Mixed-Oracles* and *Q-mixing* could be suitably combined with NeuPL, where the interaction graph would become a lower-diagonal matrix and the *Q-mixing* procedure could be similarly employed to construct best-responses to mixture-strategies. Empirically, *Mixed-Oracles* demonstrated improved rate of convergence and we agree with the reviewer that the implication of Bayes-optimality (exploration vs exploitation) in the context of population learning is in itself an interesting phenomenon that we should understand better as a field.
> > > >
> > > > We will revisit the "Related Work" section of the manuscript and adjust our writing accordingly to reflect this view. We thank the reviewer again for their continued help in improving the quality of this submission.
> > > >
> > > > **Update:** we have now updated our discussions on "Mixed-Oracles" in the Related Work section of our revised manuscript.  We hope our revised discussion would be more constructive towards future works in this direction.

---

> > > > > ### Comment · Reviewer_TfQw · 2021-11-21
> > > > > **Response**
> > > > >
> > > > > Thank you for making the requested changes. I have correspondingly increased my rating.

---

### Official Review · Reviewer_kTPT · 2021-10-31

**Correctness:** 3
**Technical Novelty And Significance:** 3
**Empirical Novelty And Significance:** 3
**Recommendation:** 8
**Confidence:** 4

**Main Review:**

**Strengths**:
1. The idea of representing an entire population of policies within a single conditional model to solve the shortcomings of existing population-based training algorithms is very elegant and novel.
2. The proposed NeuPL framework is very general and can realize a lot of current mainstream population-based training algorithms.
3. The ablation study is very detailed, which clearly demonstrate NeuPL’s effectiveness.

**Weaknesses**:
1. Although this submission is relatively well written, I think it may not be accessible to a wider range of audience since it contains too many concepts without proper explanation in the main text. For example, **a)** in the introduction section, the authors mentioned many concepts, such as game-of-skill, which should be difficult for readers who don't know much about this field to understand before reading the **game-of-skill hypothesis** proposed by [1]. **b)** In the preliminaries section, I think Figure 8 in the appendix is very helpful to understand some core concepts. I suggest putting this figure in the main text and using it as concrete examples to help readers understand these definitions. **c)** In section 1.2, the authors should introduce PSRO in more detail (perhaps in the appendix), because it is difficult for readers to understand NeuPL without PSRO. **d)** In the fourth line of the fourth page, is $E_{\sigma_i, \sigma_ j}$ a typo? I think it should be $E_{\Pi_\theta (\cdot| \sigma_i), \Pi_\theta (\cdot| \sigma_j)}$. **e)** It is better to redraw all the figures in the experiment section, all the current figures look blurry.
2. Some important related work is missing, such as [2]. Pipeline-PSRO [3] and [2] are also to accelerate the convergence speed of PSRO. It is best for the author to discuss some differences and connections between NeuPL and these methods. If possible, it is better to add some comparative experiments with [2] and [3] since the baseline PSRO-C is relatively weak.

[1] Czarnecki, Wojciech M., Gauthier Gidel, Brendan Tracey, Karl Tuyls, Shayegan Omidshafiei, David Balduzzi, and Max Jaderberg. "Real World Games Look Like Spinning Tops." Advances in Neural Information Processing Systems 33 (2020).

[2] Smith, Max, Thomas Anthony, and Michael Wellman. "Iterative Empirical Game Solving via Single Policy Best Response." In International Conference on Learning Representations. 2020.

[3] Mcaleer, Stephen, J. B. Lanier, Roy Fox, and Pierre Baldi. "Pipeline PSRO: A Scalable Approach for Finding Approximate Nash Equilibria in Large Games." Advances in Neural Information Processing Systems 33 (2020): 20238-20248.

**Summary Of The Paper:**

This submission provides an integrated and versatile NeuPL framework to improve the performance and convergence speed of population-based training algorithms by representing the entire population of policies within a single conditional policy network. NeuPL is very general, the commonly used population-based training algorithms, such as self-play, fictitious play, and PSRO can be regrade as its special cases by changing the meta-graph solver. The authors conduct extensive ablation experiments on some small games to validate the effectiveness of NeuPL from different perspectives. The results on the large-scale football environment also demonstrate NeuPL’s generalization ability.

**Summary Of The Review:**

The authors provide a framework called NeuPL to improve the performance and convergence speed of population-based training algorithms. The authors also conduct extensive experiments to validate the effectiveness and generalization ability of NeuPL. Although some important related work is missing and the writing can be further improved, this paper does solve some key problems in population-based training algorithms. Therefore, I recommend its acceptance.

---

> ### Author Response · Authors · 2021-11-19
> **Updated comments to reviewer kTPT**
>
> We thank the reviewer for recognising the significance and generality of our proposal. We are particularly grateful for the reviewer’s constructive feedback on making our submission more approachable and we have revised our writing accordingly. We now address specific questions relating to prior works.
>
> **R3/Q1: could the authors discuss some differences and connections of NeuPL to related prior works [1, 2]?**
>
> We have now updated the "Related Work" section to include a detailed discussion between NeuPL and these prior works.
>
> **P2SRO [1]** offers a principled approach to make use of additional computation resources to scale up PSRO, with concurrent learners training iterative best-responses in a staggered fashion.
>
> * P2SRO and NeuPL both suggest concurrent training of iterative best-responses as a way to scale up PSRO though NeuPL does so via an integrated multi-task learning approach, using the compute infrastructure identical to that of self-play.
>
> * P2SRO imposes an explicit ordering between policies whereas NeuPL does not by itself impose such hierarchy. In particular, NeuPL can be applied more broadly to meta-graph solvers that are not grounded (e.g. “Strategic Cycle” as in Figure 2) which is beyond the scope of P(2)SRO.
>
> * P2SRO relies on pre-defined truncation criteria for the lowest active policy that depend on the rate of marginal, empirical improvement observed. NeuPL avoids the assumption that one can efficiently distinguish between "good"-responses that have temporarily plateaued from best-responses.
>
> * In P2SRO, repeated skill learning is still needed, albeit “asynchronously” at additional costs. By contrast, NeuPL relies on direct transfer across policies, alleviate the need to learn strategy-agnostic skills repeatedly. As the opponents become sophisticated, warm-starting may not be sufficient either, as shown in Figure 5 (red).
>
> **Mixed-Oracles [2]** resembles a "mixture-of-expert" system where each policy best-responds to its predecessor **alone** (meta-game pure-strategies). To obtain a best-response to a mixture strategies, the authors used *Q-mixing* which is an approximation based on weighted Q-values in each state.
>
> * “Mixed-Oracles” and NeuPL both propose to transfer knowledge across iterations but differ in their implementation. "Mixed-Oracles" accumulates knowledge via an ensemble of previous policies. In contrast, NeuPL accumulates such knowledge directly via shared parameters in a multi-task learning regime.
>
> * “Mixed-Oracles” relies on the quality of approximation of Q-mixing which has well-understood properties in a single-state but less so in the sequential setting. In particular, such a mixture-of-expert system may not learn to to proactively gather information through interaction to reduce uncertainty, a key feature of Bayes-optimal policies. In contrast, NeuPL benefits from knowledge transfer across iterations while retaining the Bayes-optimality objectives via meta-RL training [3-4]. Our visualisation shows that when the opponent's identity is uncertain, our learned policies act proactively so as to reduce such uncertainty (e.g. the “observe-and-exploit” policy in `running-with-scissors`).
>
> We are looking into implementing P2SRO as an additional comparative study for completeness. Unfortunately, a satisfying implementation of Mixed-Oracle remains more challenging within the available time frame.
>
> **A note on “Mixed-Opponent” [2]**
>
> “Mixed-Opponent” turns the problem around and motivates an alternative interpretation of meta-game mixture-strategy: in particular, the meta-game mixture strategy is not mixed in terms of their policies on an episodic basis, but in terms of their action-value functions in the sense of Q-mixing. This is a very interesting perspective especially as the authors showed an improved rate of convergence to robust populations. Nevertheless, it remains less clear what such a mixed-opponent induces in a game theoretic sense. This warrants a separate discussion orthogonal to our main focus in this work.
>
>
>
> [1] Mcaleer, Stephen, J. B. Lanier, Roy Fox, and Pierre Baldi. "Pipeline PSRO: A Scalable Approach for Finding Approximate Nash Equilibria in Large Games." Advances in Neural Information Processing Systems 33 (2020): 20238-20248.
>
> [2] Smith, Max, Thomas Anthony, and Michael Wellman. "Iterative Empirical Game Solving via Single Policy Best Response." In International Conference on Learning Representations. 2020.
>
> [3] Ortega, P. A., Wang, J. X., Rowland, M., Genewein, T., Kurth-Nelson, Z., Pascanu, R., ... & Legg, S. (2019). Meta-learning of sequential strategies. arXiv preprint arXiv:1905.03030.
>
> [4] Mikulik, V., Delétang, G., McGrath, T., Genewein, T., Martic, M., Legg, S., & Ortega, P. A. (2020). Meta-trained agents implement Bayes-optimal agents. arXiv preprint arXiv:2010.11223.

---

### Official Review · Reviewer_xb7h · 2021-11-01

**Correctness:** 3
**Technical Novelty And Significance:** 3
**Empirical Novelty And Significance:** 3
**Recommendation:** 8
**Confidence:** 3

**Main Review:**

**Strengths**
1. The paper is written well and easy to follow.
2. NeuPL is interesting, and it also converges to an N-step best response. I also like the idea of using conditional network to represent the population of policies. It could enable skill transfer and speed up the learning.
3. The empirical study on MuJoCo Football is appreciated especially for population learning. But I recommend the authors should include more details for other researchers to reproduce the results.


**Concerns/Questions**
1. How efficient is the LP Nash solver in Algorithm 3? Will it be the compute bottleneck when $N$ is large?
2. It is not clear to me that how the conditional policy can be used in *execution* since it makes $\sigma$ as input.

**Minor comments**
1. Some figures are too blur to read.
2. Some terms are not clear to me, for example, "tabula rasa".
3. In Figure 8 (the right column), the matrix and figure do not match?


**Summary Of The Paper:**

The paper proposes Neural Population Learning which I think extends PSRO in two aspects. First, it avoid the premature *good*-response. Second, it uses a conditional network to represent the population of policies, so as to enable skill transfer.  NeuPL also offers convergence guarantees under some assumptions. NeuPL is empirically verified in *rock-paper-scissors*, *running-with-scissors*, and MuJoCo Football.

**Summary Of The Review:**

I think NeuPL is new and interesting. Also, NeuPL is also rigorously studied empirically. Although I have the question above mentioned, currently I feel the strengths of the paper outweigh the weaknesses.

---

> ### Author Response · Authors · 2021-11-19
> **Updated comments to reviewer xb7h**
>
> We thank the reviewer for recognizing the significance of our work as well as the insights surfaced in our empirical studies. We have now corrected and addressed all minor comments in our revised manuscript and we answer the remaining questions below.
>
> **R2/Q1: How efficient is the LP Nash solver in Algorithm 3? Will it be the compute bottleneck when N is large?**
>
> The LP Nash solver we used is implemented using a combination of `numpy` and `scipy` components. We performed a basic sweep over a range of payoff matrices of varying sizes on a regular desktop machine. More efficient implementations of LP solvers are readily available too if necessary [1].
>
> ```
> Solving 8x8: 100 loops, best of 5: 6.44 ms per loop
> Solving 16x16: 100 loops, best of 5: 15.6 ms per loop
> Solving 32x32: 10 loops, best of 5: 32.6 ms per loop
> Solving 64x64: 10 loops, best of 5: 126 ms per loop
> Solving 128x128: 1 loops, best of 5: 404 ms per loop
> Solving 256x256: 1 loops, best of 5: 2.01 s per loop
> ```
>
> Compared to PSRO, NeuPL makes use of an analogous meta-graph solver that incurs additional calls to the meta-strategy solver such as LP Nash. We note however that this cost is amortised over many gradient updates in practice (the hyper-parameter $T$ defined in Algorithm 5; we used $T = 1,000$ gradient updates in our experiments) and should not present a noticeable bottleneck even with a much larger neural population. Scalable Nash Equilibrium solver is also a vibrant research question in itself [2]. Finally, we note that other equilibrium solvers could be used instead of LP Nash, including more scalable meta-strategy solvers such as α-rank [3, 4].
>
> **R2/Q2: It is not clear to me how the conditional policy can be used in execution since it takes σ as input.**
>
> This is an important question that is under-discussed in our initial submission. We have now included a discussion on the test-time execution of learned policies in Appendix B.6.
>
> As is the case in all population learning methods, the final result of training is a discrete set of policies. In NeuPL, this set of policies is implemented by a tuple of two elements, the final interaction graph $\Sigma = \\{ \sigma_i \\}^N_{i=1}$ and the conditional model $\Pi_\theta(\cdot|s, \sigma)$. This tuple thus jointly define a set of concrete policies $\\{ \Pi_\theta(\cdot | s, \sigma_i), \sigma_i \in \Sigma \\}$. Nevertheless, the choice of which policy to execute at test time remains an interesting open question common for all population learning methods. The trade-offs involved in this choice is beyond the scope of this work but it remains an important open question for future research.
>
> Our project website (https://neupl.github.io/demo/) offers an interactive tool where one could select a specific pair of policies’ conditioning vectors $(\sigma_i, \sigma_j)$ in the game of `running-with-scissors`. This would hopefully help clarify the execution aspect of the learned policies populating the final neural population.
>
> **R2/Q3: could the authors provide more details for others to reproduce the results?**
>
> We share the reviewer’s concerns and we plan to release a reference implementation of NeuPL and the underlying MPO agent on our project website upon publication. The same NeuPL implementation is reused across test-domains and they only differ in their network architecture which are described in Appendix B.
>
> [1] https://developers.google.com/optimization/lp/lp_example#python
>
> [2] Gemp, I., Savani, R., Lanctot, M., Bachrach, Y., Anthony, T., Everett, R., ... & Kramár, J. (2021). Sample-based Approximation of Nash
> in Large Many-Player Games via Gradient Descent. arXiv preprint arXiv:2106.01285.
>
> [3] Omidshafiei, S., Papadimitriou, C., Piliouras, G. et al. α-Rank: Multi-Agent Evaluation by Evolution. Sci Rep 9, 9937 (2019). https://doi.org/10.1038/s41598-019-45619-9
>
> [4] Muller, P., Omidshafiei, S., Rowland, M., Tuyls, K., Perolat, J., Liu, S., ... & Munos, R. (2019). A generalized training approach for multiagent learning. arXiv preprint arXiv:1909.12823.

---

> > ### Comment · Reviewer_xb7h · 2021-11-29
> > **Thanks for the responses**
> >
> > I have increased my score and recommend an acceptance.

---

### Official Review · Reviewer_bxQq · 2021-11-03

**Correctness:** 4
**Technical Novelty And Significance:** 3
**Empirical Novelty And Significance:** 3
**Recommendation:** 8
**Confidence:** 3

**Main Review:**

The main idea is to optimize a single conditional network to learn and represent diverse policies, and reduce the computational costs (which are very expensive in prior methods) to self-play.

The paper is well written and the derivations look correct to me.

The experiments are strong, the authors illustrate that NeuPL can expected results of existing population learning algorithms on classical games, the visualization of learned policy population is helpful.

I’m not sure if transferring just shared representation (encoder and memory as in Sec 2.2) can be named as skill transfer.

The effectiveness of using this method to learn exploiters is significant. But I am wondering how much computation cost is reduced compared with prior work? If would be helpful to include a comparison.

The connection to Schaul et al 2015 is interesting but somewhat vague, it would be helpful to elaborate more, do the authors mean that NeuPL can be viewed as UVFA?

The underlying RL algorithm is MPO which arguably is not the state-of-the-art RL algorithm, what are the reasons behind using it? Does NeuRL also work with other value-based algorithms?

It’s interesting that the effective population size plateaus at 12, regardless of the speciﬁed maximum capacities of the neural populations on stochastic games, did this also happen to other domains like Mujoco based environments?

**Summary Of The Paper:**

The authors propose a new framework of  population learning that optimizes a single conditional model to learn and represent multiple diverse policies in real-world games.

Experiments on Mujoco Football and strategic games demonstrate its effectiveness.

**Summary Of The Review:**

A new framework of population training is proposed, the key idea is optimizing a shared conditional network to learn and represent diverse policies and embedding self-play.

The paper is well presented with just a few confusing points.

Experimental results are strong and have many interesting empirical findings.

---

> ### Author Response · Authors · 2021-11-19
> **Updated comments to reviewer bxQq**
>
> We thank the reviewer for taking the time to review our work and for recognising the significance of our empirical results.
>
> **R1/Q1: could the transfer of shared representation alone be considered “skill transfer”?**
>
> We recognise that this terminology has been used to refer to behavioural policies reusable across different tasks. We have now updated our manuscript to disambiguate the use of the term “skill” in our work. In this work we refer to “skills'' as capabilities that are necessary for the implementation of diverse strategies, borrowed from the Game-of-Skills hypothesis of [1]. In fact, our proposed network architecture, with its strategy-agnostic shared representation, reflects the separation between transitive skill dimensions and cyclical strategic dimension, mirroring the spinning top geometry proposed in [1]. We have now included a new diagram in Figure 10 that clarifies this connection.
>
> As a concrete example, in `running-with-scissors`, a useful, transferable “skill” may correspond to “summarising past observations” so as to support sophisticated strategic decisions. Agnostic to the opponent mixture, this skill to summarise past observations in nuanced ways must surface information that is relevant for all opponent mixtures. Our ablation studies (Figure 6) further shows that this representational skill is transferable to learning a best-response to future opponents.
>
> **R1/Q2: The connection to Schaul et al 2015 is interesting but somewhat vague, it would be helpful to elaborate more, do the authors mean that NeuPL can be viewed as UVFA?**
> We agree with the reviewer that the connection needs further clarification. Our intention is to connect NeuPL to broader multi-task learning literature with different “opponents” considered as “tasks” or “goals”.
>
> Similar to UVFA, we hope to obtain a single opponent-conditioned model that learns skills relevant to perform well against strategically diverse opponents. We observe that NeuPL induces significant positive transfer across goals (opponents), echoing observations made in [3] that shows the learned representation is highly transferable.
>
> Unfortunately, we had to remove this discussion from our main submission due to space constraints but we hope our discussion here may still be interesting to future readers.
>
> **R1/Q3: The underlying RL algorithm is MPO which arguably is not the state-of-the-art RL algorithm, what are the reasons behind using it? Does NeuPL also work with other value-based algorithms?**
>
> NeuPL is agnostic to the underlying ABR operator. Although we have not explored other RL algorithms in NeuPL, we are confident that any performant RL agents could be used instead. Nevertheless, we agree with the reviewer that agents that are designed to perform well in non-stationary, multi-task settings would be an interesting future direction.
>
> The specific choice of MPO as the underlying agent is primarily due to our familiarity with the algorithm. MPO has been shown to be comparable to other agents in recent studies (e.g. [2]) which matches our own experiences.
>
> **R1/Q4: the effective population size plateaus at 12, regardless of the speciﬁed maximum capacities of the neural populations on stochastic games, did this also happen to other domains like Mujoco based environments?**
>
> As shown in Figure 8, NeuPL only discovered a short sequence of best-responses in MuJoCo Football and did not reliably represent 8 distinct policies. The payoff matrices show that the RL agent struggled to find reliable exploits beyond 2 iterations. To understand why, we visualised the behaviours of these policies on our project website (https://neupl.github.io/demo/) where we show that the second policy already learned coordinated team play, including passing, blocking and accurate scoring.
>
> Although we cannot rule out strategic cycles where improving against one entails losing to another, we think our results offer evidence that the underlying game is highly transitive. We hypothesise that this is due to the physics-based, fully-observed nature of the game where the control policy can be continuously refined and opponents are observed at all times. Investigating the effect of NeuPL in a partially-observed variant (vision-based) of MuJoCo Football game with more players would be a very interesting future direction.
>
>
> [1] Czarnecki, Wojciech M., Gauthier Gidel, Brendan Tracey, Karl Tuyls, Shayegan Omidshafiei, David Balduzzi, and Max Jaderberg. "Real World Games Look Like Spinning Tops." Advances in Neural Information Processing Systems 33 (2020).
>
> [2] Pardo, F. (2020). Tonic: A deep reinforcement learning library for fast prototyping and benchmarking. arXiv preprint arXiv:2011.07537.
>
> [3] Schaul, T., Horgan, D., Gregor, K., & Silver, D. (2015, June). Universal value function approximators. In International conference on machine learning (pp. 1312-1320). PMLR.

---

> > ### Comment · Reviewer_bxQq · 2021-11-21
> > **thanks for the reply**
> >
> > Thanks for the clarifications and revision, I will increase my score and recommend acceptance.

---

### Author Response · Authors · 2021-11-19
**Overall comment addressing common questions.**

We thank all reviewers for taking the time to review our work and offer thoughtful and constructive feedback which has greatly improved this paper. To avoid repetitive replies, we start by addressing a few of the common issues in this comment and then reply to each reviewer’s comment specifically in threads.

**R/Q1: Experimental figures are blurry and should be regenerated.**

We thank the reviewers for bringing this to our attention which is an oversight on our part. We have now updated the manuscript with either vectorized figures where possible or higher-resolution figures otherwise.

**R/Q2: how much computation cost is reduced compared to prior work, could we include a comparison to other prior work?**

This is an important question that we considered in-depth during our experimental design. Unlike single-agent RL, comparing computation cost fairly in multi-agent strategic games is much more difficult as we need to

* correctly account for the computation cost in a distributed, population learning setting with possibly concurrent learners and;
* define a principled measure of relative performance between the resulting populations through time.

Figure 7 shows our attempt at comparing the computation cost of NeuPL to variants of PSRO under favorable conditions. Our results show that NeuPL used about 30% fewer gradient updates to reach performance parity (RPP reaching 0); more importantly, NeuPL led to a significantly more robust population of policies at convergence. Since PSRO does not prescribe a truncation criteria for each best-response learning iteration [4, 6], we additionally ran a modified PSRO-C@200k experiment which not only performed twice as many gradient updates per iteration, but also transferred parameters across iterations, yet the resulting population remains exploitable by NeuPL by a wide margin.

Orthogonal to the quantitative evaluation of computation cost in the like-for-like scenario, NeuPL also eliminates a class of “hidden” costs associated with applying game-theoretic methods to deep RL in complex games (as required in [1-3]), this includes:
* the costs of running and maintaining a performant, possibly distributed parallel learning infrastructure;
* the costs of storing, serving and evaluating a possibly unbounded number of independent models throughout an experiment;

These hidden costs, though difficult to quantify, often form barriers to entry to the wider research community. By contrast, the computational infrastructure required for NeuPL is exactly that of self-play, rendering it much more approachable computationally.

Finally, prior work such as [4, 5] sought to make PSRO scalable. [4] proposed to concurrently learn K best-responses in a staggered fashion using parallel learners as a way to scale up PSRO. However, it does not by itself propose efficiency improvement per unit of computation cost. [5] relies on properties of the underlying RL algorithm (Q-learning) to implement the opponent mixture policies via Q-mixing which is an approximation to the exact opponent mixture policy recommended by the meta-strategy solver. These discrepancies make like-for-like, fair comparison of computation cost to NeuPL challenging. We have updated our manuscript to include a comparison between NeuPL and these prior works in the Related Works section. We note that the ideas proposed in [5] such as Mixed-Oracle could be suitably combined within the NeuPL framework and we are looking into implementing [4] for completeness. We discuss how NeuPL relates to [4] and [5] more specifically in **R3/Q1**.


[1] Vinyals, O., Babuschkin, I., Czarnecki, W.M. et al. Grandmaster level in StarCraft II using multi-agent reinforcement learning. Nature 575, 350–354 (2019). https://doi.org/10.1038/s41586-019-1724-z

[2] Jaderberg, M., Czarnecki, W. M., Dunning, I., Marris, L., Lever, G., Castaneda, A. G., ... & Graepel, T. (2019). Human-level performance in 3D multiplayer games with population-based reinforcement learning. Science, 364(6443), 859-865.

[3] Liu, S., Lever, G., Wang, Z., Merel, J., Eslami, S. M., Hennes, D., ... & Heess, N. (2021). From Motor Control to Team Play in Simulated Humanoid Football. arXiv preprint arXiv:2105.12196.

[4] Mcaleer, Stephen, J. B. Lanier, Roy Fox, and Pierre Baldi. "Pipeline PSRO: A Scalable Approach for Finding Approximate Nash Equilibria in Large Games." Advances in Neural Information Processing Systems 33 (2020): 20238-20248.

[5] Smith, Max, Thomas Anthony, and Michael Wellman. "Iterative Empirical Game Solving via Single Policy Best Response." In International Conference on Learning Representations. 2020.

[6] Lanctot, M., Zambaldi, V., Gruslys, A., Lazaridou, A., Tuyls, K., Pérolat, J., ... & Graepel, T. (2017). A unified game-theoretic approach to multiagent reinforcement learning. arXiv preprint arXiv:1711.00832.

---

### Decision · Program_Chairs · 2022-01-20

**Decision:**

Accept (Poster)

**Comment:**

The authors propose a new framework of population learning that optimizes a single conditional model to learn and represent multiple diverse policies in real-world games. All reviewers agree the ideas are interesting and the empirical results are strong. The meta reviewer agrees and recommends acceptance.